# Application of a new net primary production methodology: a daily to annual-scale data set for the North Sea, derived from autonomous underwater gliders and satellite Earth observation

**Benjamin R. Loveday**[1,2], **Timothy Smyth**[1], **Anıl Akpinar**[3], **Tom Hull**[4,6], **Mark E. Inall**[5], **Jan Kaiser**[6], **Bastien Y. Queste**[7], **Matt Tobermann**[5], **Charlotte A. J. Williams**[3], **and Matthew R. Palmer**[3]

[1]Plymouth Marine Laboratory, Plymouth, UK
[2]Innoflair UG, Richard-Wagner-Weg 35, Darmstadt, Germany
[3]National Oceanography Centre, Liverpool, UK
[4]Centre for Environment, Fisheries and Aquaculture Science, Lowestoft, UK
[5]Scottish Association of Marine Science, Oban, UK
[6]Centre for Ocean and Atmospheric Sciences, School of Environmental Sciences,
University of East Anglia, Norwich, UK
[7]Department of Marine Sciences, University of Gothenburg, Gothenburg, Sweden

**Correspondence:** Benjamin R. Loveday (ben.loveday@innoflair.com)

**Abstract.** Shelf seas play a key role in both the global carbon cycle and coastal marine ecosystems through the draw-down and fixing of carbon, as measured through phytoplankton net primary production (NPP). Measuring NPP in situ and extrapolating this to the local, regional, and global scale presents challenges however because of limitations with the techniques utilised (e.g. radiocarbon isotopes), data sparsity, and the inherent biogeochemical heterogeneity of coastal and open-shelf waters.

Here, we introduce a new data set generated using a technique based on the synergistic use of in situ glider profiles and satellite Earth observation measurements which can be implemented in a real-time or delayed-mode system (https://doi.org/10.5285/b58e83f0-d8f3-4a83-e053-6c86abc0bbb5; Loveday and Smyth, 2020a). We apply this system to a fleet of gliders successively deployed over a 19-month time frame in the North Sea, generating an unprecedented fine-scale time series of NPP in the region. At a large scale, this time series gives close agreement with existing satellite-based estimates of NPP for the region and previous in situ estimates. What has not been elucidated before is the high-frequency, small-scale, depth-resolved variability associated with bloom phenology, mesoscale phenomena, and mixed layer dynamics.

## 1 Introduction

Our understanding of the global ocean has been transformed over the past 2 decades by the advent of autonomous observations from gliders and floats (Chai et al., 2020; Roemmich et al., 2019; Mignot et al., 2014; Smith et al., 2011; Testor et al., 2019). Such platforms have shown the capability to probe the marine environment at increasingly fine temporal and spatial resolution at local, regional, and global scales. Measuring essential ocean variables (EOVs), such as temperature, salinity, chlorophyll-*a* fluorescence, and photosynthetically available radiation (PAR), on these scales has greatly increased our ability to probe the links between physical systems and primary productivity (Olita et al., 2017; Thomalla et al., 2015). Further, the adoption of autonomous platforms has improved the operational reach of traditional research

vessels, which are typically cost- and weather-limited and bound to a single point in space and time. Alongside this, the international Argo float programme has grown from 0 to over 4000 floats in a little over 20 years. This network now forms a critical part of the Global Ocean Observing System (GOOS), and assimilation of data from individual floats is crucial for global weather forecast models (Le Traon et al., 2019). Currently, Argo floats are operationally constrained to the deep ocean (depth > 2 km): gliders have no such constraint, although they are around a factor of 10 more expensive and require some form of piloting, rendering them less prevalent in the global ocean.

The past 20 years have also seen a revolution in space-based sensors, widely and generically termed as satellite Earth observation (SEO). Although SEO gives unprecedented global coverage, infrared and optical sensors are limited to providing data on the near surface ($< 1\,\mu\mathrm{m}$ to $\sim 10\,\mathrm{m}$; strictly, the first optical depth) and are therefore unable to resolve variability with depth of key features such as thermoclines and deep chlorophyll maxima (Gordon and Clark, 1980; Morel and Berthon, 1989; Cullen, 2015). Additionally, passive optical and infrared SEO coverage is limited by clouds blocking the surface view. Strategies to overcome this shortcoming generally involve compositing multiple images of a region, which can lead to the smearing-out of sharp boundaries separating physically and biogeochemically distinct water masses at the sub-kilometre scale to a scale of tens of kilometres, resulting in an underestimate of spatial and temporal variability (Carr et al., 2006). The coastal domain also presents specific challenges for remote sensing of ocean colour in particular. Strong scattering, associated with high sediment loads, and absorption due to non-algal material and coloured dissolved organic matter (CDOM) make chlorophyll retrievals in Case 2 waters challenging (Morel et al., 2006; IOCCG, 2000). This complexity is compounded by the effects of bottom reflectance from shallow bathymetry (e.g. Ohde and Siegel, 2001) and chlorophyll signals that may be too high to be interpreted by standard algorithms, resulting in excessive masking.

Where SEO missions excel is their ability to provide regional to global estimates of ocean state variables at rates on timescales of days to decades, the latter depending upon the maturity of the measurement time series. An example of this, and the subject of this paper, is net primary production (NPP), the carbon fixed by plants through photosynthesis: the basis of almost all terrestrial and marine food webs. NPP plays a critical role in Earth's climate system by regulating the draw-down of atmospheric carbon dioxide (Parekh et al., 2006) and the air–sea exchange of radiatively important trace gases (Nightingale et al., 2000; Wanninkhof, 1992). SEO NPP algorithms widely estimate that marine phytoplankton fix carbon at a rate of $45\text{–}50\,\mathrm{Gt\,a^{-1}}$ (Carr et al., 2006), representing approximately half of all global NPP (Field et al., 1998). In contrast, in situ measurements of NPP in the open ocean are sparse, are generally made in the more clement months of the year, and target interesting features such as upwelling zones (Joint et al., 2002) or seasonal phytoplankton blooms (Robinson et al., 2009). Furthermore, regular fixed-point sampling (Barnes et al., 2015) is difficult to extrapolate due to spatial variability.

Significant improvements in NPP estimates from SEO surface chlorophyll-$a$ concentration ([Chl-$a$]) fields are possible with simultaneous in situ chlorophyll fluorometry and PAR profiles (Jacox et al., 2015). Hemsley et al. (2015) demonstrated and validated in the North Atlantic a method for estimating NPP at high vertical and temporal resolution using glider chlorophyll fluorescence and irradiance profiles. Significantly, it used irradiance to calibrate fluorescence and, therefore, needed no in situ samples for calibration. Hemsley et al. (2015) made depth-resolved continuous estimates of NPP over a full seasonal cycle in all weathers possible.

In this paper we present a synergistic method using a combination of in situ glider (Hemsley et al., 2015) and SEO for estimating NPP at high vertical and temporal resolution. This method is translocatable to any region of the global ocean and is designed to support processing in delayed mode (DM) and operational near-real-time (NRT) mode. It allows for flexible selection of algorithms to enable and, through the incorporation of SEO data, provides a consistent output despite inconsistent glider payloads or platform types. We apply this method to a 19-month autonomous glider field campaign in the North Sea, a critical shelf sea for fisheries with multiple other environmental stressors including eutrophication (Ferreira et al., 2011), deoxygenation (Queste et al., 2016), shipping (Barry et al., 2006), and pollution (Salomons et al., 2012). We uncover the considerable regional temporal and spatial variability in NPP across this region, capturing two winter seasons, which are crucial in conditioning the system for the following spring and summer periods. We expect future analysis of this data set, the first of its kind for the region, to provide new insights into the biophysical interplay between NPP and a complex regional oceanography defined by the influences of strong tides, topography, and fronts (Miller, 2009; Huthnance, 1991). The data set is made available via the British Oceanographic Data Centre (BODC), under https://doi.org/10/fm39 (Loveday and Smyth, 2020a).

## 2   Ingested glider data

As part of the Alternative Framework to Assess Marine Ecosystem Functioning in Shelf Seas (AlterEco) project, a sustained presence of autonomous underwater gliders in the North Sea was maintained between November 2017 and May 2019. The programme aimed to keep at least two gliders in the field at all times to provide measurement redundancy and assist with data validation. All gliders had a basic instrumentation package consisting of conductivity, temperature, and depth (CTD) in order to determine vertical profiles of temperature and salinity and a Sea-Bird Scientific

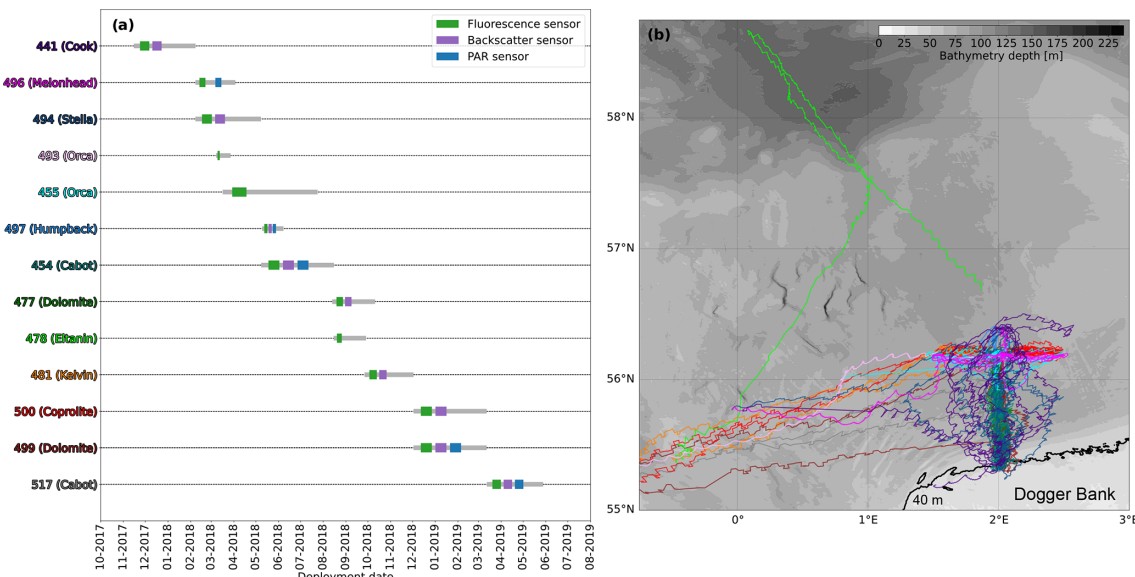

**Figure 1. (a)** Overview of the AlterEco glider deployment schedule and sensor payloads relevant to primary production calculations. **(b)** Trajectories of glider deployments overlaid on the General Bathymetric Chart of the Oceans (GEBCO) 2021 15 arcsec bathymetry for the North Sea. Track colours match the respectively coloured glider name from panel **(a)** (and with Fig. 9), with warmer track colours corresponding to later deployments. The 40 m contour, shown in black, nominally represents the outer edges of Dogger Bank.

ECO Puck for fluorescence and backscatter measurements. The data set presented here is confined to only those gliders with ECO Pucks configured for chlorophyll fluorescence measurements. Beyond this, the payload of each individual glider differed depending on the requirements of the individual mission goals (Fig. 1a). Throughout the AlterEco campaign, the gliders occupied a consistent east–west (1.5–2.5° E, along 56.1° N) and north–south (55.2–56.2° N, along 2° E) transect (see Fig. 1b), with the southern extent of the latter venturing onto Dogger Bank.

AlterEco glider missions are grouped in seven deployments[1] (Carr et al., 2019), outlined in Table 1. The glider data for these deployments are available from the BODC (https://www.bodc.ac.uk/data/bodc_database/gliders/, last access: 2 June 2022). The data are supplied in the Everyone's Gliding Observatories (EGO) format [2], which aggregates all profiles from a single glider mission into one netCDF file. More information of the spatial coverage of each of the processed missions is given in Table 2.

## 3 Method

### 3.1 Overview of the NPP processor

The NPP processor comprises a set of Python-based routines that manage the ingestion, quality control, correction, and

pre- and post-processing of autonomous underwater glider profiles as well as interfaces with external routines to calculate spectral PAR (Gregg and Carder, 1990) and NPP itself (Morel, 1991), which are implemented in the C programming language. Figure 2 shows a detailed flow diagram for the various processing stages. The processor supports multiple approaches to NPP calculation, depending on the availability of glider-based optical sensor data. Throughout this paper, when we refer to the NPP processor, we refer to the code routines that are represented in Fig. 2.

At its heart, the algorithms used to calculate NPP are as described in Hemsley et al. (2015): these in turn draw heavily upon the spectral light NPP formulation of Morel (1991). However, this method is modified to cater for fluorescence quenching and light attenuation in shelf seas, as opposed to the open ocean (as discussed in Sect. 3.4). For the purposes of determining NPP, the optimal glider instrument payload consisted of (in order of importance) chlorophyll-*a* fluorescence, PAR, and optical backscatter (Hemsley et al., 2015). Figure 1a shows that only four missions (497, 454, 499, and 517) had the full complement of required sensors, which necessitated modifications to the Hemsley et al. (2015) algorithms (see Table 1).

### 3.2 Data acquisition and staging

The processing chain was designed to accommodate either near-real-time (NRT) mode or delayed-time mode (DM) implementations and as such ingests glider data either as individual netCDF profiles as they become available (in NRT)

---

[1] available at https://doi.org/10.5285/b57d215e-065f-7f81-e053-6c86abc01a82 (Hull and Kaiser, 2020) and https://doi.org/10.5285/86429662-97b8-74fa-e053-6c86abc0a97c

[2] fully described at https://doi.org/10.25607/OBP-768

**Table 1.** An overview of mission nomenclature. All glider data used in the calculation of primary production are available at https://www.bodc.ac.uk/data/bodc_database/gliders/ and are published at the following URLs: https://www.bodc.ac.uk/data/published_data_library/catalogue/10.5285/b57d215e-065f-7f81-e053-6c86abc01a82/ and https://www.bodc.ac.uk/data/published_data_library/catalogue/10.5285/86429662-97b8-74fa-e053-6c86abc0a97c/. All URLs have been last accessed on 2 June 2022 TS1 .

| Campaign | Platform | Deployment | Glider serial | Processed here | Source data link |
|---|---|---|---|---|---|
| AlterEco1 | Fin | 439 | SG537 | No, incompatible sensors | Fin_439_R.nc |
| | Stella | 440 | unit_436 | No, early recovery | n/a |
| | Cook | 441 | unit_194 | Yes | Cook_441_R.nc |
| AlterEco2 | Orca | 493 | SG510 | Yes | Orca_493_R.nc |
| | Stella | 494 | unit_436 | Yes | Stella_494_R.nc |
| | OMG-1 | 495 | unit_352 | No, incompatible sensors | OMG-1_495_R.nc |
| | Melonhead | 496 | sg620 | Yes | Melonhead_496_R.nc |
| AlterEco3 | Cabot | 454 | unit_345 | Yes | Cabot_454_R.nc |
| | Orca | 455 | SG510 | Yes | Orca_455_R.nc |
| | Humpback | 497 | SG579 | Yes | Humpback_497_R.nc |
| | Lyra | 486 | 999 | No, incompatible sensors | n/a |
| AlterEco4 | Dolomite | 477 | unit_305 | Yes | Dolomite_477_R.nc |
| | Eltanin | 478 | SG550 | Yes | Eltanin_478_R.nc |
| | Scapa | 479 | SG602 | No, incompatible sensors | Scapa_479_R.nc |
| | Lyra | 480 | 999 | No, incompatible sensors | n/a |
| AlterEco5 | Kelvin | 481 | unit_444 | Yes | Kelvin_481_R.nc |
| AlterEco6 | Dolomite | 499 | unit_305 | Yes | Dolomite_499_R.nc |
| | Coprolite | 500 | unit_331 | Yes | Coprolite_500_R.nc |
| AlterEco7 | Ammonite | 516 | unit_304 | No, incompatible sensors | Ammonite_516_R.nc |
| | Cabot | 517 | unit_345 | Yes | Cabot_517_R.nc |
| | Scapa | 518 | SG602 | No, incompatible sensors | Scapa_518_R.nc |

n/a: not applicable

**Table 2.** Glider dive specifics per mission. The maximum profile distance is calculated from the maximum dive depth and mean dive angle and gives the maximum horizontal extent of a single dive (both down and up profiles).

| Platform | Deployment | Mean/max dive depth (m) | Mean dive angle (°) | Maximum profile distance (m) |
|---|---|---|---|---|
| Cook | 441 | 40/92 | 25 | 400 |
| Orca | 493 | 42/96 | 16 | 670 |
| Stella | 494 | 35/94 | 24 | 420 |
| Melonhead | 496 | 43/93 | 20 | 500 |
| Cabot | 454 | 33/82 | 25 | 360 |
| Orca | 455 | 44/97 | 15 | 720 |
| Humpback | 497 | 45/87 | 13 | 780 |
| Dolomite | 477 | 35/83 | 24 | 370 |
| Eltanin | 478 | 44/98 | 15 | 710 |
| Kelvin | 481 | 37/87 | 27 | 340 |
| Dolomite | 499 | 37/87 | 24 | 390 |
| Coprolite | 500 | 42/90 | 24 | 390 |
| Cabot | 517 | 33/83 | 24 | 370 |

or in EGO netCDF format (in DM). The data set described here is processed in DM. Files may be ingested locally or auto-downloaded from a remote FTP repository on a user-determined schedule. All ingested source files are stored in an initial deployment directory and catalogued in a centralised SQLite database. This non-destructive approach supports the continual updating of the glider record from a remote catalogue in the NRT case while preventing replication. The database monitors, records, and manages all subsequent stages of the processor.

NPP calculations are performed on a profile-by-profile basis. Glider data typically consist of both downward (dive) and upward (climb) components in a single file, which in our processing framework represents two profiles. If a pre-existing profile designation is provided, as is usually the case in EGO data, this is used to split the source data into profiles. If no designation is provided, the ingested data are split into single files according to the turning points in the smoothed depth record. Smoothing is performed using a fifth-order Savitzky–Golay filter, with a nominal window of 51 points. This window, which represents 5–10 min in glider sampling time, does not relate to a particular physical scale but is short enough to allow smoothing to accurately capture the tran-

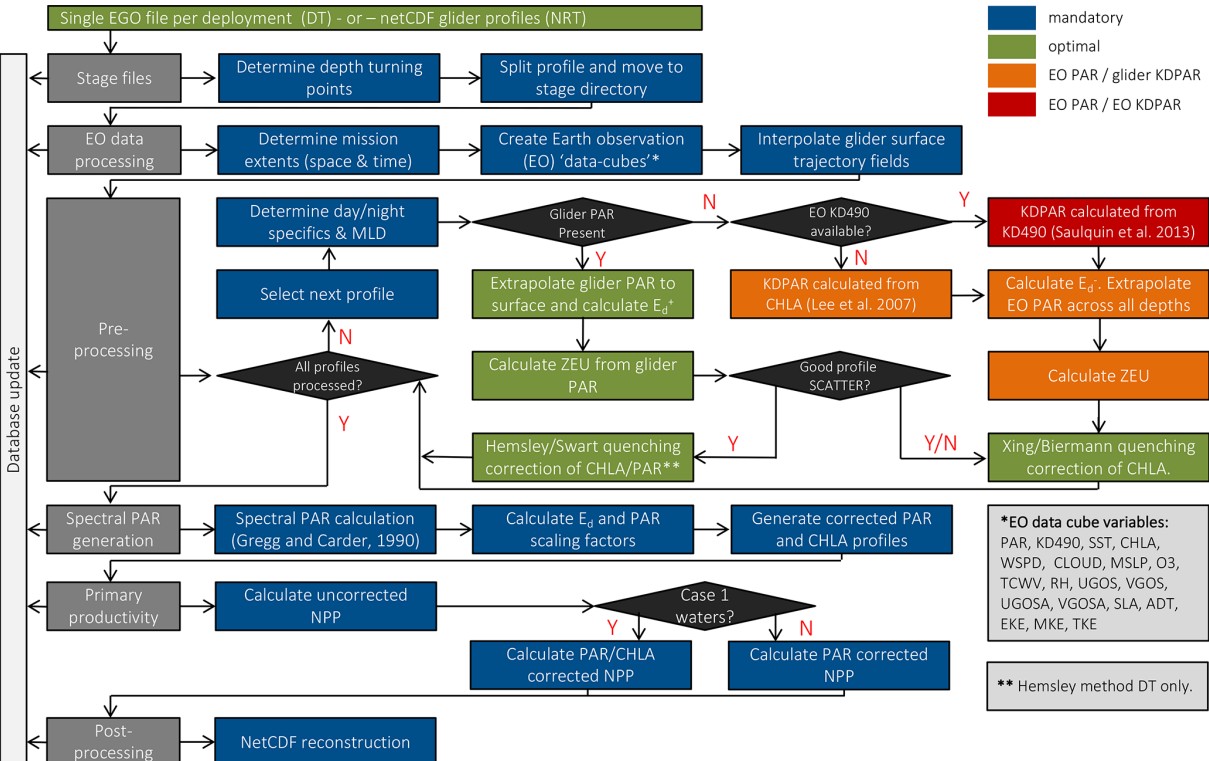

**Figure 2.** Schematic overview of the glider-based primary production processing chain. Blue boxes indicate mandatory steps; green, orange, and red boxes indicate processing options in order of decreasing preference. The light-grey inset box describes the Earth observation variables that are interpolated onto a glider's path.

sitions between descending and ascending dive components but long enough to reduce incorrect dive splitting due to short inversions or "dwelling" at the top and bottom of dives. Individual profiles are then stored in the staging directory for future use.

### 3.3 Constructing Earth observation trajectories

Due to trade-offs necessary to achieve the multiple mission priorities of the AlterEco programme, not all gliders were able to accommodate PAR sensors. Here, when required, SEO-based PAR data are used in lieu of in situ measurements (a substitution that is covered in more detail from Sect. 3.3.1 onward). This increases both the flexibility and utility of the method for the operational oceanography community, allowing it to be applied to glider data where only in situ chlorophyll-*a* fluorescence is available. In addition, SEO and reanalysis data are used to provide information on the prevailing atmospheric and marine conditions during each glider mission. A list of SEO and reanalysis data sources and the variables that are extracted and/or derived can be found in Table 3 and in Fig. 2.

When a glider mission is updated (e.g. a new profile is added in NRT mode), the processor calculates the new temporal and spatial extents of the mission. Using these extents,

the processor gathers the required SEO and reanalysis data from the specified source, concatenates the retrieved catalogue in time, and trims the spatial coverage to produce a "data cube" that matches the glider mission extents. The spatial trimming is performed remotely, on the server side, if the data service in use allows this capability, reducing data transfer costs and time. In NRT mode, new data are added to extend the cube as required, without the need to download the entire catalogue once again (e.g. via the concatenation of new time slices to the existing local record). This operation is performed for all variables and for all gliders, irrespective of whether they have the relevant in situ measurement, allowing for the continual validation of the use of SEO and reanalysis data as a substitute.

Once the data cube has been constructed, the average time and location of each profile are extracted and concatenated into a one-dimensional time series of the glider trajectory. Bilinear interpolation is then used to retrieve the corresponding SEO and reanalysis data from the relevant data cube, resulting in an SEO trajectory file for each variable, with a value for each profile. During construction, the cube is both spatially and temporally "padded" to eliminate "edge effects" associated with interpolation.

**Table 3.** List of SEO and reanalysis variables used to support glider missions. Bold type variables are derived by the primary production processor.

| Description | Provider | Source | Variables |
|---|---|---|---|
| Sea surface topography | CMEMS | SEALEVEL_GLO_PHY_L4_NRT & REP | Sea level anomaly<br>Absolute dynamic topography<br>Absolute geostrophic velocities<br>Geostrophic velocities anomalies<br>**Eddy kinetic energy**<br>**Total kinetic energy**<br>**Mean kinetic energy** |
| Atmospheric variables | ECMWF | ERA-I | 10 m wind speed $(u/v)$<br>Total cloud cover<br>Mean sea level pressure<br>$[O_3]$<br>[Water vapour]<br>2 m temperature<br>2 m dew point<br>**Wind speed**<br>**Relative humidity** |
| Optical variables | NASA | MODIS L3m Daily products | PAR<br>$K_{d490}$<br>**Instantaneous PAR** |
| Ocean tracers | CMEMS | GLOBAL_ANALYSIS_FORECAST_PHY | Sea surface temperature<br>Sea surface salinity<br>Mixed layer depth |
| Biogeochemistry | CMEMS | OCEANCOLOUR_GLO_CHL_L3_NRT & REP | [CHLA]<br>**Euphotic depth** |

### 3.3.1 Treatment of SEO PAR trajectories

SEO-based broadband PAR values ($E_d$), defined as the average PAR value between 400 and 700 nm, are derived from MODIS daily average values, measured in moles per square metre per day (Frouin et al., 1989) (see Table 3). Instantaneous values of broadband PAR, corresponding to the glider measurement times, are derived from the average daily value as follows. The light distribution is modelled as a sine curve between sunrise ($T = 0$) and sunset ($T = \pi$). The amplitude of this curve is determined such that the integrated value below it matches that of the daily average. The instantaneous value is then extracted by interpolating the value from the curve at the glider measurement time. The instantaneous PAR value is finally converted to watts per square metre.

### 3.4 Pre-processing and calibration

The pre-processing step consolidates the glider and SEO-based data on a profile-by-profile basis, performs quality control procedures, and selects the relevant variables for NPP calculations depending on availability (Fig. 2). Sporadic missing values are common in in situ data. Where possible, linear interpolation is used to fill these gaps in the positional, depth, and pressure data. If interpolation is not possible, the profile is discarded, and no further processing takes place.

Following this, and where not provided directly, conservative temperature and absolute salinity are calculated from the glider CTD record using the TEOS-10 Python GSW toolbox [3]. Mixed layer depth (MLD) is then calculated from the temperature and density gradients using a hybrid algorithm that accounts for profile shape, giving more accurate estimates than threshold-based methods that rely on a fixed value (Holte and Talley, 2009). If the MLD calculation fails (e.g. due to missing depth data), the MLD from the previous profile is used. This is only allowed once. The MLD is prevented from being shallower than 5 m as depths shallower that this are typically poorly sampled by a glider. Once the physical variables are processed, the PAR and [Chl-$a$] profiles are assessed, along with the backscatter data, if present.

PAR data delivered in raw counts are corrected to watts per square metre using the calibration coefficients specific to the sensor. The in situ sensor fluorescence data, measured in volts, are converted to chlorophyll concentration ([Chl-$a$]; mg m$^{-3}$) by multiplying by the scale factor (cali-

---

[3]https://teos-10.github.io/GSW-Python/ (last access: 2 June 2022)

bration coefficient) specific to the sensor and subtracting the manufacturer-provided dark count. Backscatter is similarly calculated.

An additional dark correction is then applied to the [Chl-*a*] measurements. As all glider data in the AlterEco programme are made available in delayed mode, the minimum value of [Chl-*a*] is extracted on a per-profile basis across the entire mission. To remove the influence of large negative outliers, the global minimum value is then calculated as 3 standard deviations less than the mean value of the time series of profile minima. This value is then subtracted from the entire record. Throughout the remaining processing, any [Chl-*a*] data $< 0.0 \, \mathrm{mg \, m^{-3}}$ are then assumed to be erroneous and are discarded.

On rare occasions, glider 497 (Humpback) recorded occasional spikes of over $10^3 \, \mathrm{mg \, m^{-3}}$ in the [Chl-*a*] data. These measurements are not considered to be reliable, and therefore all values over $10^3 \, \mathrm{mg \, m^{-3}}$ are discarded. In addition, glider 481 (Kelvin) experienced a sudden "step change" of $> 5 \, \mathrm{mg \, m^{-3}}$ in [Chl-*a*] at depths below both the MLD and $Z_{\mathrm{eu}}$ as compared with the initial deployment value toward the end of its mission. Consequently, all data for this glider after this point are discarded.

The NPP processor offers the possibility of introducing additional calibration factors based on independent in situ measurements taken at the time of glider deployment and recovery. Unfortunately, in the case of the AlterEco programme, no such measurements were taken, and so no additional calibration of the [Chl-*a*] data is performed. To ensure that the manufacturer calibration is sufficient in this case, the surface [Chl-*a*] data from each glider are compared with their SEO counterpart (Table 3). The results are shown in Fig. 3. From the figure, it is evident that, as expected, each glider shows significantly more variability than its SEO counterpart. However, for all gliders, the median value extracted from the glider is similar to its SEO counterpart. Further, with the exception of the Orca missions, the interquartile range for each glider overlaps with its SEO counterpart. This suggests that there is no significant bias in the glider [Chl-*a*] record and that the manufacturer calibration was sufficient in most cases.

Where available, optical backscatter measurements ($b_{\mathrm{bp}}$) may be used to correct the surface chlorophyll fluorescence profile for near-surface quenching (Hemsley et al., 2015). The backscatter data are initially passed through a seven-point running minimum filter to remove spikes (Thomalla et al., 2018). Negative values are removed, and the backscatter profile is subsequently interpolated onto the glider depth record.

As with the treatment of the PAR and backscatter data, the [Chl-*a*] record is interpolated onto the glider depth record on a profile-by-profile basis. On occasion, due to very short dives or quality control processes conducted on the original EGO format data, the [Chl-*a*] record is sparse to such an extent that interpolation onto the depth record is not possible.

Where this occurs, the entire profile is discarded, and no NPP calculation is performed.

### 3.4.1 Determining the PAR profile

PAR sensors do not always acquire at the same sampling rate as the glider CTD sensor. Consequently, where available, in situ PAR data for a given profile are interpolated onto the glider depth record prior to further processing. The decision point for the use of glider or SEO-based broadband PAR is made according to the prioritisation of the following three cases.

– *Case 1*. Where a profile falls during the daytime and glider $E_{\mathrm{d}}$ is available, this is used by default (though the use of SEO-PAR can be forced to permit validation). $K_{\mathrm{dPAR}}$ is calculated from the linear regression of the logged PAR values with depth. The regression is weighted by the square root of the magnitude of the logged PAR values, emphasising the effect of the surface layers. $E_{\mathrm{d}}$ at depth is then projected to the surface using the $K_{\mathrm{dPAR}}$ value, giving near-surface broadband PAR ($E_{\mathrm{o}}^{-}$).

Broadband PAR at the surface (or just above) ($E_{\mathrm{o}}^{+}$) is then derived from $E_{\mathrm{o}}^{-}$ using Eq. (2) from Hemsley et al. (2015) (Eq. 1, below). A value of 0.04 is used for the irradiance reflectance, $R$ (Victoria Hemsley, personal communication, 2018), and 0.48 for the Fresnel reflectance, $\bar{r}$. Total reflectance, $r_{\mathrm{tot}}$, the sum of the direct reflectance ($r_{\mathrm{d}}$) and diffuse reflectance ($r_{\mathrm{diff}}$), is calculated via the method specified in the supplementary material of Hemsley et al. (2015). The required wind speed is provided from the SEO trajectory files.

– *Case 2*. Where glider PAR is not available, SEO-based surface broadband PAR ($E_{\mathrm{o}}^{+}$) is substituted. $E_{\mathrm{o}}^{-}$ is then calculated by rearranging Eq. (1). The same values as above are used for the irradiance reflectance and Fresnel reflectance. SEO $\overline{K_{\mathrm{dPAR}}}$ is calculated from SEO $\overline{K_{\mathrm{d490}}}$ using the turbid water exponential model described by Eq. (9a) and (9b) of Saulquin et al. (2013). The calculated $\overline{K_{\mathrm{dPAR}}}$ is then used to project broadband PAR into the subsurface across the glider depth record.

– *Case 3*. Although derived from the same source, SEO $K_{\mathrm{d490}}$ is occasionally not available, even though PAR is. In this case, the euphotic depth is determined according to Eq. (2) (Lee et al., 2007), where CHL represents the maximum in situ [Chl-*a*] measured above the MLD. $\overline{K_{\mathrm{dPAR}}}$ is then calculated according to Eq. (3), where PAR($Z = Z_{\mathrm{eu}}$) is assumed to be 1 % of PAR($Z = 0$).

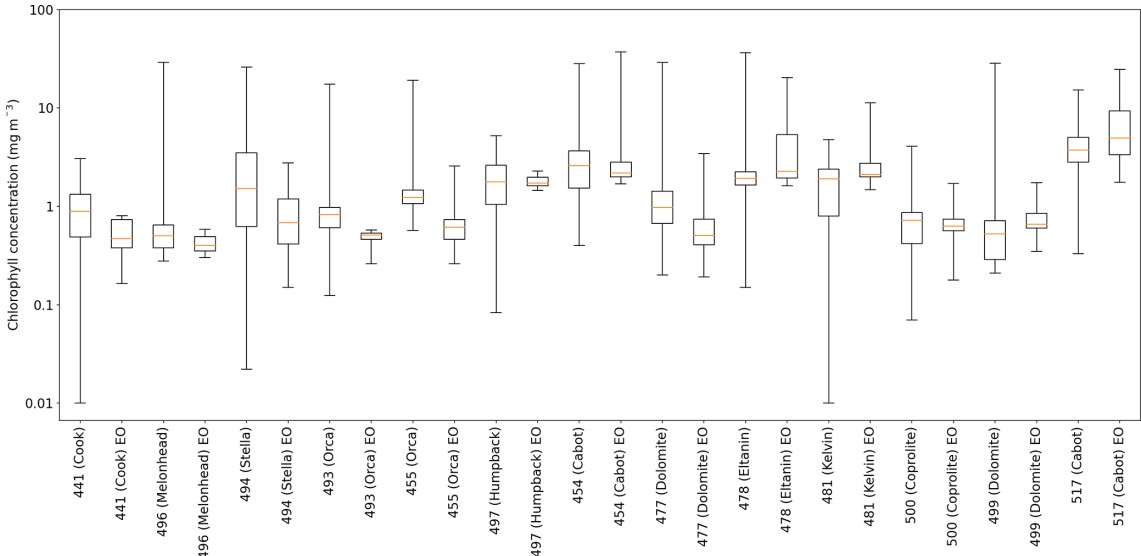

**Figure 3.** Statistical comparison of the surface chlorophyll values measured by each glider with its SEO trajectory counterpart. From bottom to top, the values shown on the box-and-whisker plot are the minimum, lower quartile range (25 %), median, upper quartile range (75 %), and maximum. Only the glider measurements with an SEO counterpart are considered.

$$E(0)^+ = \frac{E(0)^-(1 - R\bar{r})}{(1 - r_{\text{tot}})} \tag{1}$$

$$Z_{\text{eu}}/(\text{m}) = 34.0 \times \text{CHL}^{-0.39}/(\text{mg m}^{-3}) \tag{2}$$

$$\overline{K_{\text{dPAR}}} = \frac{(\ln(E_d(Z = 0)) - \ln(E_d(Z = Z_{\text{eu}})))}{Z_{\text{eu}}} \tag{3}$$

The PAR record is labelled as bad and is not processed (i) in the case of night-time profiles; (ii) where [Chl-$a$] data could not be interpolated (see Sect. 3.4); or (iii) where the glider is within 5 m of the bathymetry depth, as interpolated from the GEBCO 15 arcsec gridded product[4]. The latter criteria prevent the glider from deriving NPP estimates from readings that may have been gathered at depths where particle resuspension is likely to make the PAR estimates derived from SEO unreliable, given that we assume a constant value of $K_{\text{dPAR}}$.

The calculation of euphotic depth ($Z_{\text{eu}}$), a necessary parameter in some quenching algorithms, is dependent on the case being used. Under Case 1, $Z_{\text{eu}}$ is defined as the depth at which the light level is 1 % of the surface value. Under Case 2, $Z_{\text{eu}}$ is calculated from $K_{\text{dPAR}}$. Under Case 3, $Z_{\text{eu}}$ is calculated from Eq. (2). $Z_{\text{eu}}$ is calculated for all good profiles. The case used is stored in the EUPHOTIC_DEPTH_FLAG variable of the final data set (please see Table 4).

To validate this approach, Fig. 4a and b compare the in situ (red) and SEO-based $E_o^+$ (blue) estimates for gliders 517

(Cabot) and 454 (Cabot), respectively. The SEO-based interpolation method gives an accurate facsimile of the daily PAR cycle, with a mean $\bar{E}_o^+$ that falls within 7 % of the in situ value (an error value that is comparable with the 5 % "in-air" performance of the in situ PAR sensor itself [5]). However, it has a notably lower standard deviation. This is somewhat expected as the SEO-based values do not take account of the instantaneous cloud conditions.

Across both missions, the SEO surface PAR underpredicts the surface PAR reconstructed from the glider profiles. At solar noon, the nominal peak in the daily PAR value, this equates to an average anomaly of $\sim 50\,\text{W m}^{-2}$. However, the comparison of instantaneous values is problematic and overstates the discrepancy between the two time series. The daily integrated PAR time series align much more closely, with mean values from the glider (solid black line) of $5268\,\text{kJ m}^{-2}$ for 517 (Cabot) and $6265\,\text{kJ m}^{-2}$ for 454 (Cabot) and from SEO (dashed black line) of $5041\,\text{kJ m}^{-2}$ for 517 (Cabot) and $6224\,\text{kJ m}^{-2}$ for 454 (Cabot), respectively.

### 3.4.2 Quenching correction of the chlorophyll fluorescence profile

Fluorescence quenching in phytoplankton is caused by a variety of physiological acclimation mechanisms in order to avoid photodamage under excessive irradiance (Kiefer, 1973). This effect typically manifests as a depression of the fluorescence signal in the surface waters during daylight and

---

[4]https://www.gebco.net/data_and_products/gridded_bathymetry_data/ (last access: 2 June 2022)

[5]https://www.seabird.com/asset-get.download.jsa?id=54627862114, (last access: 2 June 2022)

**Table 4.** Variables present in the EGO format netCDF data files. All variables have a single "time" dimension.

| Variable name | Quantity | Units |
|---|---|---|
| TIME | Time | Seconds since 1970-01-01 |
| PROFILE_NUMBER | Glider profile number | None |
| LONGITUDE | Longitude | Degrees east |
| LATITUDE | Latitude | Degrees north |
| PRESSURE | Pressure | Decibar |
| DEPTH | Glider depth | m |
| CHLA | Quenching-corrected [CHL-$a$] | $\mathrm{mg\,m^{-3}}$ |
| MIXED_LAYER_DEPTH | Mixed layer depth | m |
| EUPHOTIC_DEPTH | Euphotic depth (ZEU) | m |
| EUPHOTIC_DEPTH_FLAG | Euphotic depth method flag | None |
| DOWNWELLING_PAR | Photosynthetically active radiation (PAR) | $\mathrm{W\,m^{-2}}$ |
| DOWNWELLING_PAR_FLAG | PAR method flag | None |
| DOWNWELLING_PAR_EO | PAR from satellite Earth observation (SEO) | $\mathrm{W\,m^{-2}}$ |
| DOWNWELLING_PAR_EO_FLAG | SEO PAR method flag | None |
| PRIMARY_PRODUCTION | Primary production (PP) from in situ PAR | Carbon flux of $\mathrm{mg\,m^{-3}\,d^{-1}}$ |
| PRIMARY_PRODUCTION_EO | PP from SEO PAR | Carbon flux of $\mathrm{mg\,m^{-3}\,d^{-1}}$ |
| DEPTH_INTEGRATED_PRIMARY_PRODUCTION | PP integrated to ZEU | Carbon flux of $\mathrm{mg\,m^{-2}\,d^{-1}}$ |
| DEPTH.........\_PRODUCTION_EO | SEO PP integrated to ZEU | Carbon flux of $\mathrm{mg\,m^{-2}\,d^{-1}}$ |

Note: DOWNWELLING_PAR_FLAG and DOWNWELLING_PAR_EO_FLAG are equivalent but are included twice as they are relevant to both of their associated variables.

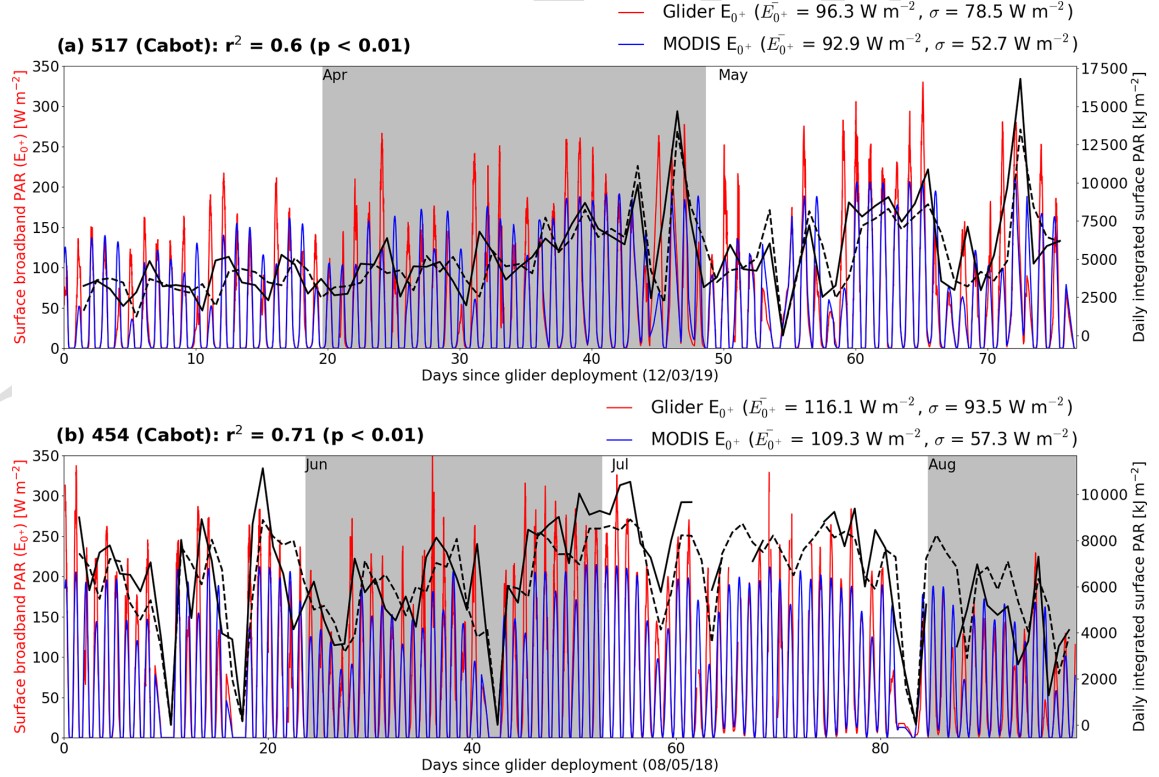

**Figure 4.** Comparison of glider- and MODIS-based surface broadband PAR ($E0^+$) values. Red and blue traces show the 10-profile smoothed PAR for the glider and interpolated MODIS products, respectively. These are recorded against the left-hand axis. All statistics ($r^2$, $\mu$, $\sigma$) are based on the valid, unsmoothed time series data for daytime profiles only. The solid and dashed black traces, measured against the right-hand axis, show the daily integrated PAR values for the glider and MODIS products, respectively, and are derived from the corresponding red and blue traces.

particularly around solar noon, when the downwelling irradiance is at a maximum (Xing et al., 2012; Biermann et al., 2015). Multiple approaches to quenching correction have been proposed (e.g. Xing et al., 2012; Biermann et al., 2015; Hemsley et al., 2015; Swart et al., 2015; and Thomalla et al., 2018). The applicability of these methods depends on the region being studied and the availability of optical backscatter data. Four methods are tested for this data set:

- the Xing et al. (2012) method, where the maximum [Chl-*a*] measured in the mixed layer is projected to the surface for daytime profiles (this method can be used in either DM or NRT cases);

- the Biermann et al. (2015) method, where the maximum [Chl-*a*] measured above the euphotic depth is projected to the surface for daytime profiles (this method can be used in either DM or NRT cases);

- the Swart et al. (2015) method, where the optical backscatter signal above the euphotic depth is used to correct the corresponding [Chl-*a*] on a profile-by-profile basis (this method can be used in either DM or NRT cases);

- the Hemsley et al. (2015) method, where, again, the optical backscatter signal is used to correct the corresponding [Chl-*a*] using the night-time relationship with backscatter, as measured across the entire glider mission (this method can be used in DM cases only).

Due to the lack of available light during night-time sampling, [Chl-*a*] profiles remain unquenched. The extensive variability in shelf seas makes direct correction of daytime profiles to their nearest night-time counterpart challenging (Carberry et al., 2019). However, when quenching is appropriately accounted for daytime [Chl-*a*] profiles should, in the aggregate, approximate their night-time counterparts. Figure 5 compares the histogram distribution of night-time and daytime [Chl-*a*] profiles for four tested methods across the entire missions of gliders 517 (Cabot) and 454 (Cabot). Optical complexity in coastal waters, associated with the presence of sediment, undermines the relationships between [Chl-*a*] and the backscatter record. Consequently, while it may perform well in the open ocean, the quenching correction method described by Hemsley et al. (2015) performs poorly in this case. The Swart et al. (2015) method is shown to be similarly unsuitable for the same reason.

The Xing et al. (2012) method clearly outperforms the other methods tested and is used to process all the gliders deployed during the AlterEco programme. Its strong performance is ascribed to its ability to appropriately capture the regional seasonal interplay between the MLD and euphotic depth in the shelf seas. As shown in Fig. 6, the MLD sits above the euphotic depth during spring. This allows for the establishment of a deep chlorophyll maximum (DCM) (see

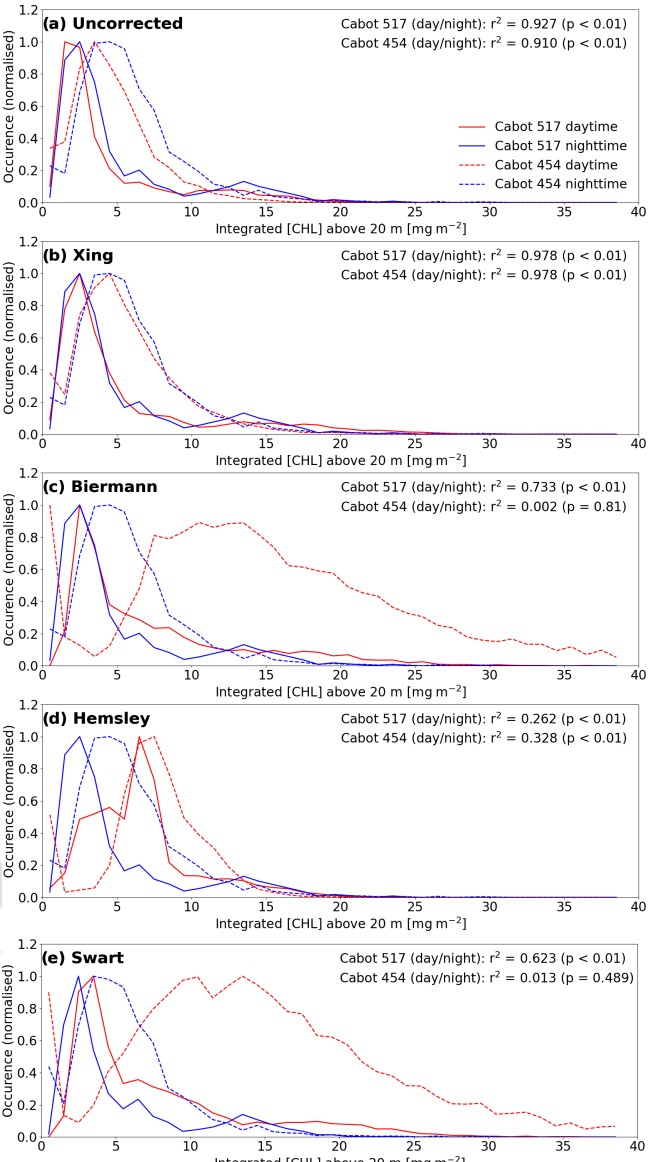

**Figure 5.** The effects of the implementation of different quenching mechanisms on the distribution of the integrated [Chl-*a*] in the top 20 m for Cabot deployments 454 (dashed lines) and 517 (solid lines). Panel **(a)** shows the distribution of daytime (red) and night-time (blue) [Chl-*a*] in the uncorrected case. Panels **(b)**, **(c)**, **(d)**, and **(e)** show the effects of implementing the mixed-layer-depth-based correction of Xing et al. (2012), euphotic-depth-based correction of Biermann et al. (2015), and the backscatter-based corrections of Hemsley et al. (2015) and Swart et al. (2015).

Fig. 7), which is particularly important for NPP in this region (Fernand et al., 2013). In this case, quenching corrections using euphotic depth as a maximum depth limit (e.g. Biermann et al., 2015) overcorrect as they tend to encapsulate the DCM in the quenching correction process, extrapolating erroneously high [Chl-*a*] to the surface. This is understandable as these approaches were indeed designed to account for

sub-surface chlorophyll maxima, but in open-ocean regions, where the MLD is typically deeper than the $Z_{eu}$.

## 3.5 Calculating and scaling the spectral irradiance profile

Once the pre-processing stages have been completed for all available glider profiles (Fig. 2), spectral $E_d$ profiles are calculated for each glider profile using the solar irradiance model described by Gregg and Carder (1990). To account for local meteorological conditions, the model runs using the total column atmospheric ozone ($[O_3]$), cloud cover, wind speed, relative humidity, and total column water vapour parameters for each profile are stored in the relevant trajectory file (Table 3). These spectral $E_d$ values calculated from the model are scaled such that their integrated value between 400 and 700 nm matches the corresponding $E_o^+$ measurements provided by the glider or SEO data sources (see Sect. 3.4.1). This scaling correction accounts for instantaneous sky conditions associated with each profile.

## 3.6 Implementing chlorophyll-*a* scaling

The work by Hemsley et al. (2015) implemented a novel methodology to exploit the relationship between PAR and [Chl-*a*] to account for changes in the apparent fluorescence to chlorophyll calibration, brought about by phytoplankton community succession. This approach allows dynamic changes to the calibration and reduces the need for in-field calibration, which is difficult, if not impossible to implement, especially in the near-real-time case. However, this method is based on an in-water model suitable for Case 1 waters (Carr, 1986), with the non-water component of light attenuation ascribed to [Chl-*a*] only (Morel and Maritorena, 2001).

The processor retains the ability to implement this method, as detailed extensively in Hemsley et al. (2015) and represented in Fig. 2 by the "Case 1" decision box. However, the optically complex waters of the shelf seas are rich in sediment and do not conform to the Case 1 paradigm. Implementation of a spectral irradiance model more suitable to the region requires the consistent deployment of in situ PAR and backscatter sensors that is not available across the AlterEco programme. Consequently, no PAR-based scaling of the [Chl-*a*] profiles is performed for this data set. This caveat is further discussed in Sect. 5.4.

## 3.7 Calculating NPP

Net primary production, $P$, is calculated from the corrected [Chl-*a*] and spectral downwelling PAR profiles using the Morel (1991) model, as presented in Hemsley et al. (2015) and shown in Eq. (4). The model calculates NPP through a triple integral across day length ($L$), depth ($D_1 = 0$, $D_2 = Z_{eu}$), and wavelength ($\lambda_1 = 400$ nm, $\lambda_2 = 700$ nm). The absorption cross section per unit of chlorophyll ($a^*$; $m^2\,g^{-1}$)

and net growth rate ($\phi_\mu$; mol(carbon) mol(quanta)$^{-1}$]) are parameterised as in (Morel, 1991).

$$P = 12\,\mathrm{g\,mol^{-1}\,d^{-1}} \int_0^L \int_{D_1}^{D_2} \int_{\lambda_1}^{\lambda_2} [\mathrm{Chl}-a](Z)E_d(t,Z,\lambda)a^*$$

$$(\lambda)\phi_\mu(t,Z,\lambda)\mathrm{d}\lambda\mathrm{d}Z\mathrm{d}t \qquad (4)$$

NPP estimates, in units of carbon flux ($\mathrm{mg\,m^{-2}\,d^{-1}}$), are calculated for all corrected [Chl-*a*] profiles using the per-profile average time and position for each. The piecewise measurements are integrated from the 1 % light level (as determined by the model) to the surface to give a final estimate of depth-integrated primary productivity in carbon flux of milligrams per square metre per day.

Figure 8 allows a comparison of using SEO-based PAR in the calculation of spectral $E_d$ and subsequent NPP in contrast to using in situ PAR. SEO-based PAR is shown to function as a suitable proxy in this method, remaining highly correlated with its in situ counterpart, with mean values that are within 2 % of the target estimate.

When combined, the NPP times series derived from the AlterEco glider deployments spans a 19-month period, as shown in Fig. 9. As expected, NPP is at its greatest in the spring and early summer and reaches its highest in the spring of 2019, corresponding with the timing of the regional spring bloom. Conversely, it drops to near zero in the winter months, when light availability becomes limiting. The figure also shows the inherent spatial and temporal variability in the time series, reflected in the inter-glider and intra-glider data, respectively. Despite operating during the same period, glider 454 (Cabot) measures approximately twice the NPP of glider 455 (Orca) throughout April, May, and June 2018, where we expect biological activity to be near its highest level.

Black traces, indicating NPP estimates derived from in situ PAR sensors, compare well with their coloured (SEO-derived) counterparts in all cases. However, the divergence between the signals recorded by the concurrently deployed gliders 455 (Orca), 497 (Humpback), and 454 (Cabot) strongly suggests the presence of significant variability in the region north of the Dogger Bank (Fig. 1b, lower-right corner).

## 4 Data provenance and structure

The complete finalised data set consists of 13 netCDF files, in EGO format. Each netCDF file corresponds to a single glider mission. The data cover a region spanning a latitude of 51.005 to 58.669° N, a longitude of −1.497° W to 2.577° E, and a time period of 15 November 2017 to 28 May 2019. During deployment 481 (Kelvin) the glider remained at the surface from 21 November to 2 December 2018 and did not acquire [Chl-*a*] data, and so no NPP was calculated for this period. No other glider was deployed during this time, resulting in a single 10 d gap in the record. Each EGO data

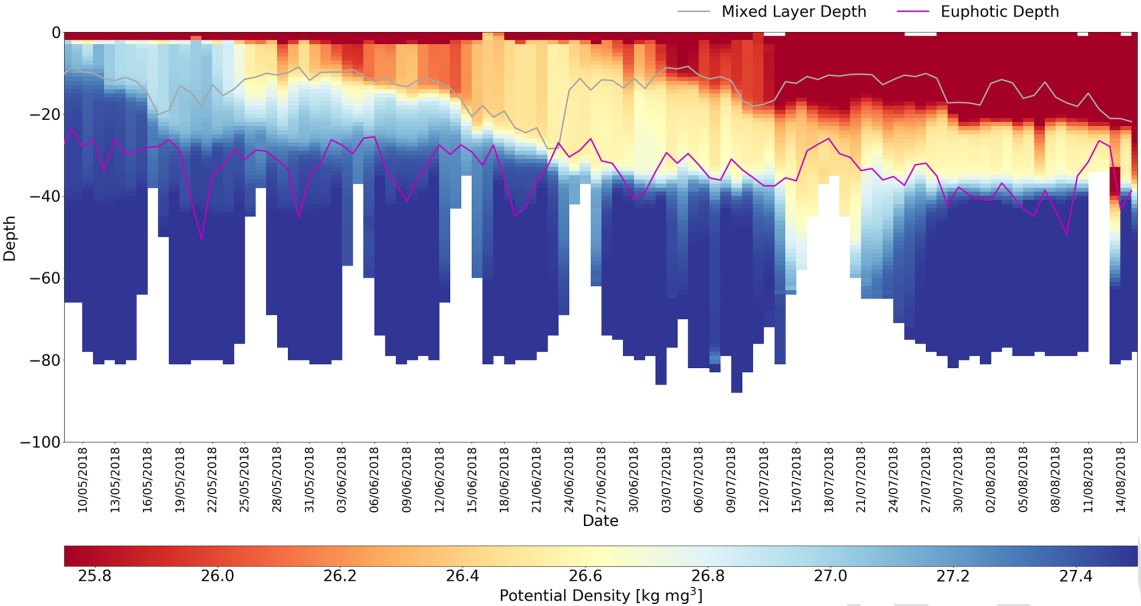

**Figure 6.** Potential density distribution for time series for 454 (Cabot). Mixed layer depth and euphotic depths are shown by the respective grey and purple traces, which have been smoothed using a 10-profile window. Gaps in the euphotic depth time series correspond to night-time profiles.

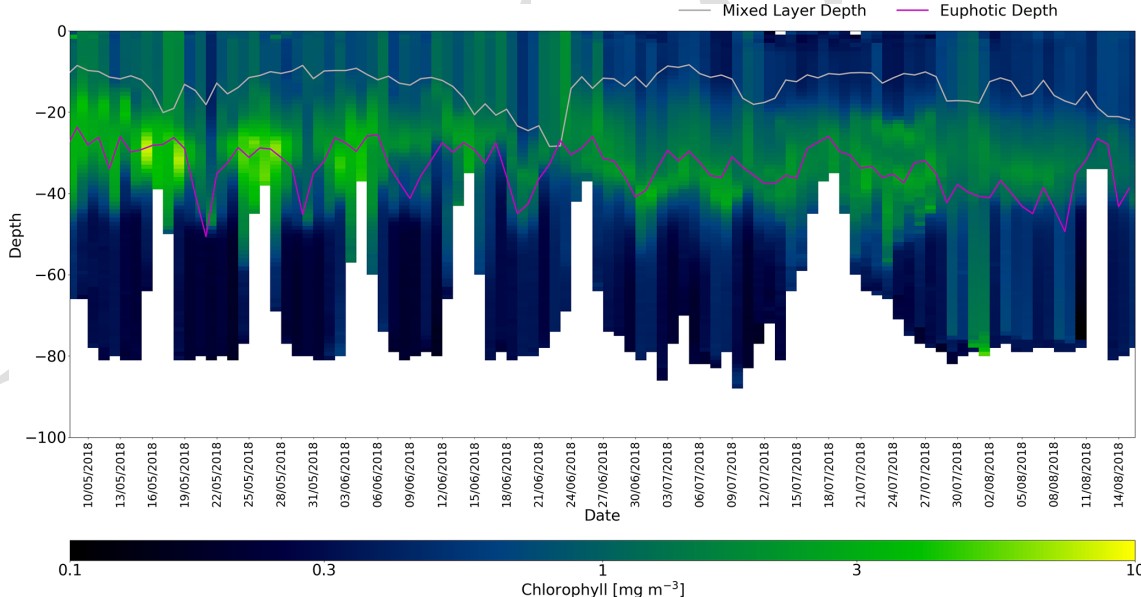

**Figure 7.** Quenching-corrected [Chl-$a$] time series for 454 (Cabot). The Xing et al. (2012) quenching correction is applied. Mixed layer depth and euphotic depths are as in Fig. 6).

file contains the variables listed in Table 4. The intermediate variables calculated as part of the processor are not included in the netCDF data files. However, we intend to publish the NPP processor in full, allowing future users to make use of it and adapt the methodology to their own purposes. More information on the availability of the code can be found in the "Code and data availability" section

of this paper. It is important to note that, to avoid duplication, each netCDF output file does not include the temperature and salinity variables used in the NPP processor. However, these can be found via BODC at the following link https://www.bodc.ac.uk/data/bodc_database/gliders/ (last access: 2 June 2022) and have a one-to-one mapping to the NPP

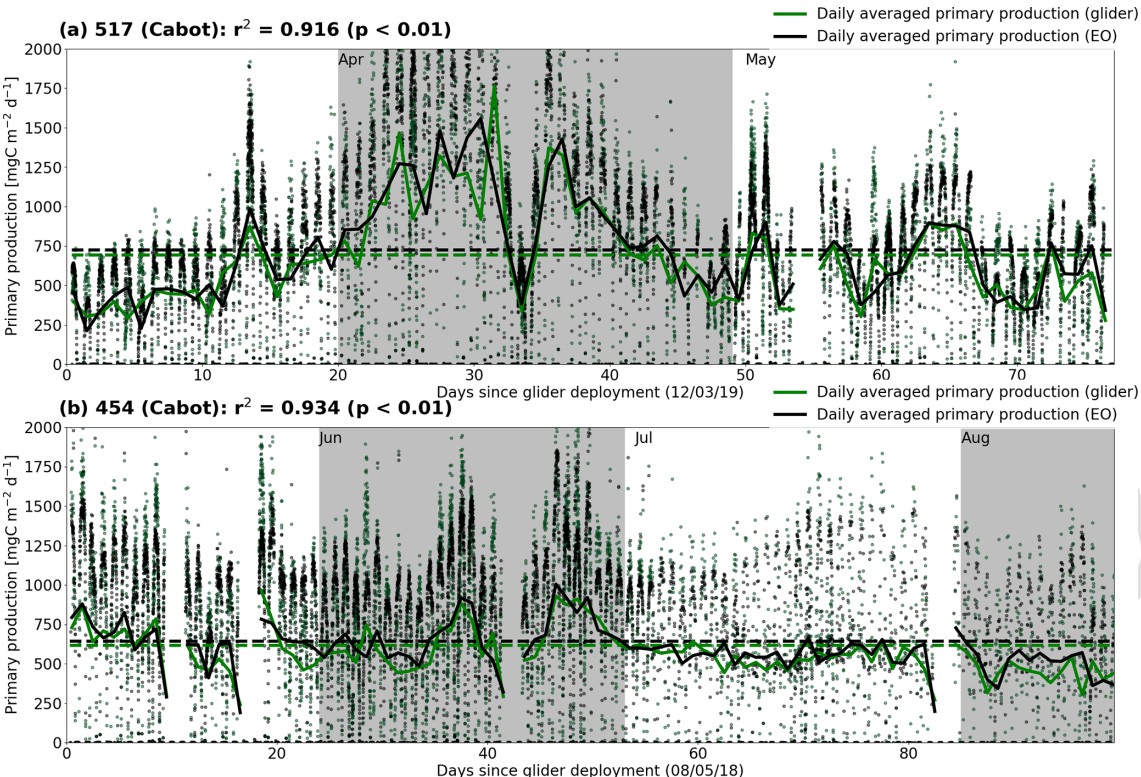

**Figure 8.** Comparison of net primary production estimates from the in situ PAR method (green points and traces) and SEO-based PAR method (black points and traces) for glider Cabot missions **(a)** 517 and **(b)** 454. Points represent the instantaneous measurements taken from individual profiles, with solid traces showing the daily integrated values. Total mean daily values for each mission and method are given by the respective dashed lines. The $r^2$ statistic is calculated between the individual profile values for the two methods.

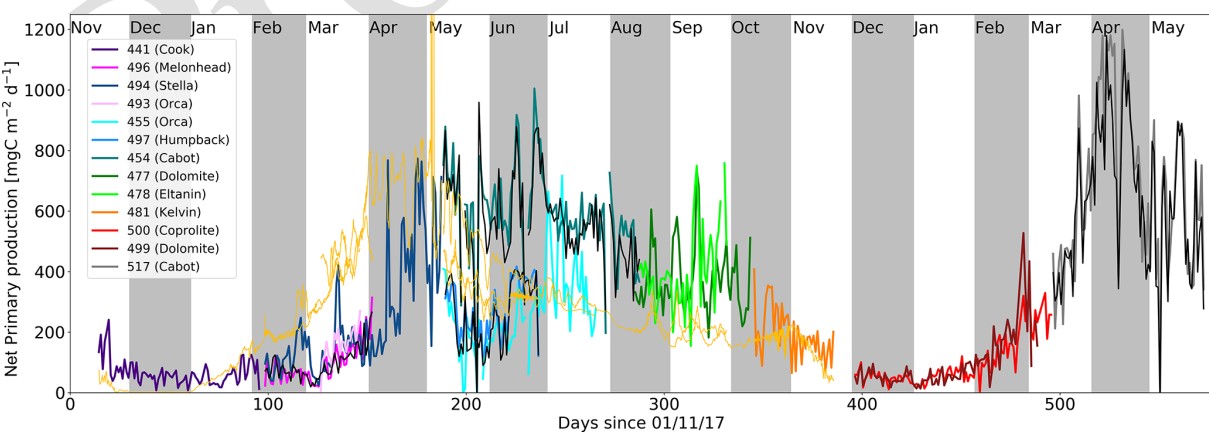

**Figure 9.** Daily column-integrated net primary production rate estimates from all gliders deployed in the AlterEco programme (see Table 1). Coloured traces represent each glider mission (matching those used in Fig. 1) and show the net primary production estimates derived from the SEO-based PAR method. Where in situ PAR sensors were available, a corresponding black trace for that glider mission is also shown. The thin orange trace shows the net primary production extracted from v4.2 of the ESA Ocean Colour Climate Change Initiative (OC-CCI) NPP product for each glider where available.

glider data set. The specific links for each glider are included in the final column of Table 1.

Responsibility for maintaining the data set lies with Plymouth Marine Laboratory, the provenance authority for the final output. No updates of the data set are expected. The data set is stored in the British Oceanographic Data Centre (BODC) archive and has the following digital object identifier: https://doi.org/10.5285/b58e83f0-d8f3-4a83-e053-6c86abc0bbb5 (Loveday and Smyth, 2020a).

## 5 Data validity

### 5.1 Fluorescence quenching validity

In order to quantify the efficacy of the various quenching correction algorithms upon the chlorophyll fluorescence profiles, a comparison was made between the daytime (quenched) and night-time (not quenched) [Chl-$a$] over the top 20 m of the water column. This comparison assumes that there is little change in the vertical profile of [Chl-$a$] in a 24 h period: this is obviously a simplification as there will be changes due to (1) bloom growth or loss (e.g. via respiration, decay, and grazing) and (2) spatial variability. Figure 5 shows that the major discrepancy between the night-time and the uncorrected daytime [Chl-$a$] profiles for glider 454 (Cabot; see Fig. 1 and Table 1) is when the integrated [Chl-$a$] in the top 20 m exceeds 4 mg m$^{-2}$ and is less than $\sim$ 12 mg m$^{-2}$. The percentage variance between daytime and night-time uncorrected cases is 91.0 % with a significant $p$ value of $< 0.01$. The Xing et al. (2012) method (Fig. 5) clearly outperforms the other correction methods tested in this case ($r^2 = 97.8\%$, $p < 0.01$; cf. Hemsley et al., 2015: $r^2 = 32.8\%$, $p < 0.01$; cf. Biermann et al., 2015: $r^2 = 0.2\%$, $p = 0.81$). Similar results were obtained for glider 517 (Cabot; see Fig. 1 and Table 1), with Xing et al. (2012) ($r^2 = 97.8\%$, $p < 0.01$) performing best, followed by Biermann et al. (2015) ($r^2 = 73.3\%$, $p < 0.01$) and Hemsley et al. (2015) ($r^2 = 26.2\%$, $p < 0.01$).

Figure 7 shows the corrected [Chl-$a$] profiles for glider 454 (Cabot). The corrected time series shows a gradual deepening of the DCM over the 3-month mission from around 25 m in mid-May (2018) to 40 m in mid-August. During this period there is also a clear reduction in the peak [Chl-$a$]: $> 3$ mg m$^{-3}$ at the start of the mission, $\sim 1$ mg m$^{-3}$ towards the end. The transects on and off Dogger Bank (depths shoaling to around 40 m) are clearly correlated with a shoaling of the euphotic depths from $\sim 30$ m on the bank to $\sim 50$ m off the bank. Changes in the MLD are less pronounced, apart from a deepening between 18 and 22 June 2018 and a rapid shoaling from 30 to 18 m around 24 June 2018. Meteorologically, June 2018 was characterised by a relatively slack pressure gradient (light winds) until a brief 3 d period of stronger north or north-westerly winds on 19 and 22 June, which corresponds to the episodic deepening of the MLD in this period.

### 5.2 Comparison with historical measurements in the North Sea

Two distinct advantages of gliders are that they sample flexibly, in terms of horizontal space and depth, and they can gather data at high frequency. As there are no pre-existing measurements of NPP in the North Sea with comparable frequency, here we compare our results with available estimates of annual mean productivity from both satellite and in situ sources.

Independent [Chl-$a$] estimates, derived from v4.2 of the ESA Ocean Colour Climate Change Initiative (OC-CCI) data set (Sathyendranath et al., 2019), indicate that the 2018 spring bloom was relatively intense in comparison to 2019, a fact that does not appear to be reflected in the glider NPP data. Kulk et al. (2020) applied the NPP methodology of Platt and Sathyendranath (1988) and Sathyendranath et al. (2020) to this satellite [Chl-$a$] record, producing a 20-year time series of satellite-based NPP from 1998–2018. Although this record does not span the entire AlterEco period (2017–2019), is only calculated at 9 km resolution, and is only available as a monthly product, it provides a useful data source to compare the glider NPP measurements against. Satellite-based NPP estimates for each glider are shown in light orange in Fig. 9. Comparing the glider and co-located satellite time series further suggests that 494 (Stella) likely failed to fully capture the onset of the spring bloom in 2018. However, given the disparity between the estimates obtained by 454 (Cabot), 494 (Stella), and 497 (Humpback), it is likely that the region is subject to significant spatial heterogeneity, which perhaps the satellite product is too coarse to record.

Table 5 summarises the monthly and annual NPP estimates across all AlterEco glider campaigns. The monthly mean and standard deviations derived from the satellite NPP record, calculated across a box spanning the AlterEco sampling region, are shown in the final two columns of Table 5. The annual cycle and mean annual NPP rate as measured from the glider missions agree well with contemporaneous values interpolated from the monthly mean OC-CCI NPP record. It is notable from the table that, while still within 1 standard deviation, the April NPP peak in the glider data is somewhat lower than its OC-CCI counterpart, likely due to the low signal recorded by 494 (Stella) over this period in 2019. However, the glider-based NPP signal peaks at a time consistent with remote sensing estimates.

The glider-based annual mean NPP value is 98 g C m$^{-2}$ a$^{-1}$. This compares favourably with the 119 g C m$^{-2}$ a$^{-1}$ measured by Joint and Pomroy (1993), who applied a $^{14}$C approach to measure daily NPP through extensive surveys carried out over ICES Region 7 (north of Dogger Bank), scaling up to monthly estimates using the mean daily value across the region and number of days per month. It also compares well to the annual estimate of 125 g C m$^{-2}$ a$^{-1}$ for the northern North Sea proposed by van Beusekom and Diel-Christiansen (1994), based on

**Table 5.** Monthly statistics for SEO-PAR-based depth-integrated primary production estimates across all glider missions. Values for in situ PAR-based depth-integrated primary production estimates are given in brackets where available. All measurements are given in carbon flux, measured in grams per square metre per day, unless otherwise specified. The final column gives the mean primary production extracted from v4.2 of the monthly OC-CCI climatology from 1 January 1998 to 31 December 2018 over a box spanning the core of the AlterEco sampling region (55.25–56.25° N, 1.5–2.5° E).

| Month | NPP mean | NPP SD | $N$ profiles | OC-CCI NPP mean* | OC-CCI NPP SD* |
|---|---|---|---|---|---|
| January | 53 | 21 | 8190 | 64 | 69 |
| February | 138 (70) | 139 (23) | 9925 (2381) | 166 | 31 |
| March | 192 (184) | 125 (131) | 10 868 (5105) | 358 | 79 |
| April | 470 (607) | 269 (251) | 7393 (3336) | 617 | 112 |
| May | 391 (416) | 157 (158) | 9382 (7132) | 594 | 131 |
| June | 406 (476) | 110 (142) | 7951 (5916) | 340 | 73 |
| July | 364 (397) | 127 (138) | 1634 (1498) | 303 | 49 |
| August | 334 (344) | 83 (74) | 3928 (628) | 278 | 58 |
| September | 317 | 91 | 4165 | 234 | 51 |
| October | 192 | 73 | 2923 | 177 | 31 |
| November | 134 | 50 | 2554 | 109 | 55 |
| December | 49 | 16 | 5276 | 4 | 13 |
| Annual | 269 | 199 | 74 189 | 270 | 63 |
| Annual ($mg\,m^{-2}\,d^{-1}$) | 269 | 199 | 74 189 | 270 | 63 |
| Annual ($g\,m^{-2}\,a^{-1}$) | 98 | 73 | 74 189 | 99 | 23 |

* Data provided by Plymouth Marine Laboratory, based on Kulk et al. (2020).

a synthesis of daily NPP estimates from multiple cruises. The glider measurements are similarly consistent with NPP estimates derived from models, with Varela et al. (1995) recording $130\,g\,C\,m^{-2}\,a^{-1}$ for ICES Region 7 (as used by Joint and Pomroy, 1993), Moll (1998) simulating $119\,g\,C\,m^{-2}\,a^{-1}$ across the northern North Sea, and Zhao et al. (2019) reporting $82.6$–$118.8\,g\,C\,m^{-2}\,a^{-1}$ for the central and northern North Sea in their tidal simulations.

Alongside NPP, Varela et al. (1995) provide estimates of gross primary production (GPP). In the northern North Sea (ICES Region 4), an NPP of $149\,g\,C\,m^{-2}\,a^{-1}$ is associated with a GPP of $314\,g\,C\,m^{-2}\,a^{-1}$. Assuming that the ratio of NPP : GPP remains broadly constant in the region on an annual basis, we can apply this to our glider NPP measurements to obtain a GPP estimate of an approximate value of $\sim 200\,g\,C\,m^{-2}\,a^{-1}$. This compares favourably with the measurements of Capuzzo et al. (2018), who reported an annual mean gross production of $200 \pm 15\,g\,C\,m^{-2}\,a^{-1}$ in seasonally stratified regions from 1998 to 2013 (including Dogger Bank).

## 5.3 Value and utility

Primary production is highly variable on short temporal and spatial scales. The impact of the mesoscale variability associated with fronts (Olita et al., 2017; Taylor and Ferrari, 2011) and eddies (Hansen et al., 2010; Hu et al., 2014) can be extensive. High-frequency changes in tidal phase (Zhao et al., 2019), sky conditions, and the local wave field (Reed et al., 2011) can also exert a strong influence. To monitor the impact of these processes in highly productive shelf seas, it is desirable to continually sample key regions using technologies that support adaptive sampling strategies. Autonomous underwater vehicles (AUVs), such as gliders, offer one such approach to this problem, offering persistent monitoring of shelf sea biogeochemistry (Chai et al., 2020; Liblik et al., 2016) and informing regional model assimilation strategies (Skákala et al., 2021).

This data set presents the first intra-annual, glider-based in situ NPP time series for the North Sea that is able to address questions pertaining to biophysical interactions on a high-frequency basis. From Fig. 9 it is clear that the NPP signal is modulated at multiple frequencies within individual deployments, and substantial spatial heterogeneity exists between co-deployments (e.g 455 (Orca) and 454 (Cabot)). Further analysis of this data set should give insight into the physical processes that contribute to this variability.

When deployed with multiple mission goals in mind, glider payload space typically comes at a premium. Most notable in this case is the effect on the deployment of PAR sensors, which are present on less than 50 % of missions. However, adaptation of previous methodology to accommodate SEO-based PAR estimates has been shown to be feasible. Combining SEO surface data with AUV profiles also presents interesting options for reconstructing subsurface fields. Machine learning methods have demonstrated the feasibility of combining SEO surface fields with in situ profiles to render a three-dimensional picture of ocean biogeo-

chemical properties (Sauzède et al., 2015, 2016). The data set presented here would be well suited for application of such methods to evaluate and further extend coverage of NPP data in the global ocean.

The processing method developed here allows for glider-based NPP to be calculated in a much broader array of cases. While in DM it can replicate the approach of Hemsley et al. (2015), the inclusion of differing quenching algorithms promotes application to different regions and/or different sensor loads (e.g. those without backscatter). The flexible inclusion of SEO data in lieu of in situ PAR measurements expands this utility even further, allowing NPP calculations from gliders with a more limited array of sensors without substantial loss of accuracy (Fig. 8). Finally, the ability to support NRT ingestion of glider data allows for NPP calculation in an operational setting.

## 5.4 Limitations, scope, and future improvements

PAR, when spectrally decomposed, can be used to provide a calibration of the [Chl-*a*] fluorometer (Hemsley et al., 2015). Although the fluorometer calibration may be accurate at the start of an individual mission, calibration using nearby discrete [Chl-*a*] samples at launch and retrieval of the glider may lead to a false sense of security, particularly in areas of high heterogeneity, such as experienced during this study. Hemsley et al. (2015) showed that within-mission variability in the correction factor is possible due to changes in phytoplankton community structure. However, the model previously proposed is suitable for Case 1 waters only and does not account for absorption and scattering by CDOM and sediment, respectively, and so no dynamic calibration is applied here. The strong agreement between AlterEco glider NPP measurements and both satellite and historical in situ estimates (see Sect. 5.2) underlines the validity of the data set, and future work will consider the incorporation of a model to cater for more complex waters, where glider payload allows.

As noted in Sect. 3.4, the measurement of in situ dark counts for fluorescence is performed on the entire glider mission. This method is therefore inappropriate for near-real-time analysis of glider profiles. Inclusion of a methodology to calculate dark counts for both the fluorescence and backscatter measurements on a per-profile basis, such as that developed by Wojtasiewicz et al. (2018), would also be advantageous.

While the quenching correction method of Xing et al. (2012) proved most appropriate in this case, this result should not be considered a general solution. This rationale underpins the decision to incorporate multiple methods to correct near-surface fluorescence; however, the method eventually chosen is limited by the sensors deployed and most notably the availability of backscatter data (Fig. 1). The availability of backscatter data allows for a wider selection of correction methodologies in both DM processing (Hemsley et al., 2015) and NRT processing (Swart et al., 2015). In addition, its in-

clusion is essential to constructing a complex water model, as discussed earlier in this section. As the NPP processor was developed during the AlterEco programme, which commenced in 2017, it only takes advantage of quenching methods available at the time. Future work is expected to include more recent quenching methodologies such as Thomalla et al. (2018).

For long-duration missions (i.e. more than a few days) bio-fouling of sensors mounted on gliders can affect data quality. Unlike Argo floats, which typically park at depths well below the euphotic zone ($\sim 1$ km), for 10 d, gliders spend a greater portion of their time in the photic zone, allowing the build-up of a bacterial substrate and then algal colonisation. Despite many strategies to mitigate bio-fouling (copper-covered sensors, bio-wipers), it is impossible to completely eradicate it currently, and even predicting its onset is problematic. Anecdotally on moorings situated in the western English Channel (Smyth et al., 2010a, b), bio-fouling has been observed to take several months to colonise sensors and then following cleaning has only taken a few weeks to re-emerge. Best efforts have been made to truncate the glider [Chl-*a*] record where bio-fouling appears evident (Sect. 3.4.2).

Here, the methodology described is used to generate a primary productivity data set in an optically complex shelf region. However, much of the basis of the methodology is derived from previous work that was developed for use in the open-ocean context (e.g. Hemsley et al., 2015). Consequently, we expect the NPP processor to be viable in the open ocean, where chlorophyll concentration tends to dominate the optical signal. In the open ocean, quenching methods based on calibration against the backscatter record are also likely to perform better (e.g. Swart et al., 2015) and, in the case of Hemsley et al. (2015), allow for dynamic calibration associated with changes in phytoplankton community structure. As Earth-observation-based retrieval of chlorophyll concentration typically has lower errors in the open ocean, there may be opportunities to investigate the use of remotely sensed data to correct and dynamically calibrate the in situ chlorophyll record, an approach previously suggested by Lavigne et al. (2012). It is, however, important to point out that the methodology may require tuning when used in different mission contexts. With deeper and/or longer dives, care should be given to select the correct smoothing parameters to determine the turning points of the profile. In addition, where in situ PAR is not available, it may also be advisable to select a $\overline{K_{\mathrm{dPAR}}}$ model that is more suited to clear waters, such as Morel et al. (2007). More broadly, future investigations should also consider the effect that the choice of $\overline{K_{\mathrm{dPAR}}}$ model used has on the resulting NPP value.

## 6 Code and data availability

The data are made available via the British Oceanographic Data Centre (BODC), via https://doi.org/10/fm39 (Loveday

and Smyth, 2020a). Their use may be cited using Loveday and Smyth (2020a). Access to the code for the primary productivity processor will shortly be made available via https://github.com/timjsmyth/GliderPP (Loveday and Smyth, 2020b) TS2.

## 7 Conclusions

This paper discusses the generation of a 19-month, near-continuous glider-based data set of net primary production in the North Sea, the first of its kind for the region. The methodology used to derive this time series is discussed in detail, with a specific focus on the approaches taken to account for fluorescence quenching and the use of SEO-based PAR data in lieu of in situ sensors. While, in this case, pre-processed glider data from the AlterEco programme serve as a starting point, consideration is also given to adaptation of the method for NRT and operational use. Although limitations in the approach used are discussed, especially in regard to the feasibility of dynamic calibration and effects of bio-fouling, the results show strong agreement with previous studies as well as satellite-derived estimates and the results of biogeo-chemical model simulations. They present a unique, depth-resolved picture of the high-frequency variability and spatial heterogeneity present in the rates of NPP for the region and highlight the advantages of using autonomous systems to persistently monitor the shelf seas, especially in tandem with remote-sensing-based approaches. The newly developed processing approach also has implications for the development of a PP indicator (e.g. through the Marine Strategic Framework Directive food web descriptor), overcoming some of the temporal and spatial sampling limitations that have historically undermined its inclusion in assessments, relegating its listing to candidate only.

**Author contributions.** BRL and TS developed the methodological approach and led writing of the manuscript. BRL built the processing system and generated the resulting data set. The remaining authors are responsible for the glider deployments, the provision of supporting data sets and calibration information, and providing methodological input in the manuscript.

**Competing interests.** The contact author has declared that neither they nor their co-authors have any competing interests.

**Disclaimer.** Publisher's note: Copernicus Publications remains neutral with regard to jurisdictional claims in published maps and institutional affiliations.

**Acknowledgements.** This work was conducted under the NERC Alternative Framework to Assess Marine Ecosystem Functioning in Shelf Seas (AlterEco) programme. The authors thank Emma Slater for managing data ingestion and hosting at BODC, Thomas Jackson at Plymouth Marine Laboratory for providing the OC-CCI data used in Table 5 and Fig. 9, and Hayley Evers-King for proofreading and comments on the manuscript narrative. We are also grateful to our two anonymous reviewers and especially Sandy Thomalla for their insightful comments and suggestions for improvements.

**Financial support.** This research has been supported by UK Research and Innovation (grant nos. NE/P013910/1, NE/P013899/1, and NE/P013902/1).

**Review statement.** This paper was edited by François G. Schmitt and reviewed by Sandy Thomalla and two anonymous referees.

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

## Remarks from the typesetter