# Peer review of "Application of a new net primary production methodology; a daily to annual-scale data set for the North Sea, derived from autonomous underwater gliders and satellite Earth observation."

_Earth System Science Data, 2021_

## Author Comment (AC1)

**ESSD response to Anonymous Referee #1**
09/02/2022

**Overview:**

The paper presents a data set of net primary production (PP) concentrated in an area of 2 degrees lat x 2 degrees lon, with data also from a wider 4 degrees lat x 5 degrees lon area, in North Sea covering a period of 19 months from November 2017 to May 2019. The data derive from the integration of in situ, satellite and glider observations. Therefore the calibrated data derive form the single point measurements along the obliquous glider trajectories, which is what is normally done using glider data. The procedure for obtaining depth integrated primary production values, which is one of the final products, is similar to that by Hemsley et al, 2015, cited by the authors, which is turn based on different bio- optical models to obtain: $E_d^+$, $E_d(z,\lambda)$, and PP itself.

1. So far I could not access to the data base, because I always got 'service unavailable', therefore I don't know if any technical annex is available.

   *We apologise to the reviewer for this. We believe that the issue is related to the splitting of the following URL between lines during typesetting:*

   *https://www.bodc.ac.uk/data/published_data_library/catalogue/10.5285/b58e83f0-d8f3-4a83-e053-6c86abc0bbb5/)*

   *We have corrected this in the paper by providing the DOI link, as opposed to the full URL.*

2. It would be useful to add some technical details, such as the descent angle and the depth range of glider oscillation. I assumed a descent angle of 28° which, for a 0-1000 me excursion, would mean a spatial span of ~400 m for each down-up trajectory.

   *A new table, table 2 has been added. This contains the requested information on the specifics of each dive mission.*

3. I am a little perplex, not having any spectral data, of using a spectral model for PP. I am aware that it was used also by Hemsley et al, but it looks like a sort of vicious circle. I am wondering if, considering the errors inherent in each of the used models, how the results would compare with those of a simpler, non- spectral model. However the effort may pave the way to the use of spectral sensor which might be available soon on ARGO profilers. Another possible analysis would be to compare their PP results with the backscattering derived POM, for the gliders that are equipped with the sensor.

   *We appreciate the reviewers point and agree that a methodological comparison between spectral and non-spectral primary production models would be of interest to both the glider and ARGO communities. However, comparing our results with the output from other models is beyond the scope of this analysis and beyond the remit of ESSD papers.*

4. My other perplexity is about publishing this data set on ESSD. Not too much for the limited spatio-temporal span but because it seems to me that the main contribution of the study is that of having implemented a flexible, automatic procedure which may be used in many other contexts. This procedure, for what I read is made available by the authors. A Journal like Enviornmental Science and Technology could be, in my opinion, more appropriate. However, I think that this is mostly an editorial choice.

*We thank the reviewer for this comment. However, while we acknowledge that the methodology developed to product this data set receives significant attention in the manuscript (a point noted by other reviewers), we still consider the main thrust to be the data set itself as it is unprecedented for the region. As such, we maintain that this paper falls with ESSD's editorial scope.*

---

## Author Comment (AC2)

**Overview:**

This paper by Loveday et al presents a method to estimate in situ net primary production based on automous gliders deployed for 17 months during the AlterECO project, and the subsequent data set.

I found the manuscript well written and clearly explained explained. In particular I appreciated the discussion about the different methods to correct non-photochemical quenching, their comparison and the discussion about the reason why a method clearly outperfoms the other ones.

I believe this data set is relevant for the community and important as well from a methodological point of view. Therefore I recommend the manuscript to be published in Earth System Science Data after some minor revisions.

**Comments:**

Without being an expert of the field, I am reflecting on the relevance of such a method for the open-ocean domain. I would like this question to be tackled in the manuscript in order to provide a generalizing view of the method. Some work on existing autonomous plaforms (Argo floats, glider) exist (eg Lavigne et al., 2012) to correct in situ fluorescence with surface satellite chlorophyll-a measurements. Could such an approach be used to contraint the glider based chlorophyll-a data? Would it be beneficial in the present case? Would another method for correcting the quenching perform better than the one used in the present paper?

(ref : Lavigne, et al. "Towards a merged satellite and in situ fluorescence ocean chlorophyll product." Biogeosciences 9.6 (2012): 2111-2125.)

> *As requested, we have expanded on the "Limitations, scope and future improvements" section of the discussion to include the points raised above, including reference to Lavigne, et al. (2012). Please see the final point in this response for more information.*

**Technical corrections:**

l14 : To introduce the glider technology, please cite the community paper of Testor, et al. "OceanGliders: a component of the integrated GOOS." Frontiers in Marine Science 6 (2019): 422.

> *This reference has been added.*

l99 : Is the 151 points correspond to a particular physical scale?

> *Firstly, we apologise to the reviewer, as there is a typographic error here. The smoothing window should read as 51 points, not 151. We have corrected this. Secondly, no, the number is not relevant to a general physical scale, but to this glider data set specifically. Changes to the glider sampling rate, such as when used in deeper waters or during longer missions, is likely to influence this number. The ingested data set was tested with a range of smoothing windows. A window of 51 points resulted in the most successful division of profiles. This value, (which, for reviewe'rs interest, represents approximately 5-10 minutes of glider sampling time) is small enough to allow smoothing to accurately capture the transitions between descending and ascending dive components, but long enough to reduce incorrect dive splitting due to short*

*inversions or "dwelling" at the top and bottom of dives. We have made a small addition to the paper to reflect this paragraph.*

l154 : (Saulquin et al., 2013)

*This has been corrected.*

l158 : (Lee et al. 2007)

*This has been corrected.*

l354 : while discussing limitation of the method, the case of offshore waters could be also discussed regarding the application of a similar approach to compute NPP estimates, and the required potential tuning of the method.

*The following paragraph has been included at the end of the newly renamed "Limitations, scope and future improvements" section, in response to the point made above and in the comments;*

*Here, the methodology described is used to generate a primary productivity data set in an optically complex shelf-region. However, much of the basis of the methodology is derived from previous work that was developed for use in the open ocean context (e.g. Hemsley et al., 2015). Consequently, we expect that the NPP processor to be viable in the open ocean, where chlorophyll concentration tends to dominate the optical signal. In the open ocean, quenching methods based on calibration against the backscatter record are also likely to perform better, and, in the case of Hemsley et al., 2015, allow for dynamic calibration associated with changes in phytoplankton community structure. In addition, as Earth observation-based retrieval of chlorophyll concentration typically has lower errors in the open ocean it raises the prospect of using remotely sensed data to correct, and dynamically calibrate the in situ chlorophyll record, an approach previously suggested by Lavigne et al., (2012). It is important to point out that the methodology may require tuning when used in different mission contexts. With deeper and/or longer dives, care should be given to select the correct smoothing parameters to determine the turning points of the profile.*

---

## Author Comment (AC3)

**ESSD response to Dr. Sandy Thomalla, reviewer 3**
09/02/2022

**Overview:**

This paper presents a multi year (19 month) dataset of glider and complimentary ocean colour net primary production (NPP) data with a variety of complementary ancillary products for a study in the North Atlantic. The data set in and of itself is new (although similar approaches have been implemented elsewhere to produce comparative data sets in support of physical-biogeochemical research). I found the supporting article to be appropriate and well articulated. Although I was unable to access the data itself, from the accompanying text it appears that the data set is of high quality with appropriate metadata and consideration of constraints and limitations. It is however not clear to me whether all steps in the data processing are retained in the data set to allow for a user to edit one or more of the authors steps in their post processing if need be e.g. to implement a different quenching correction method or an adjusted Kd PAR calculation or whatever the case may be. I believe that this data set meets the ESSD criteria of producing a data set that can be further reused for the benefit of Earth system science research, in particular by providing additional insight into that characteristics of regional, seasonal and sub-seasonal variability in the biological response to physical drivers at high resolution. That being said, to do so effectively would, I think, require the accompanying Temperature and Salinity profile data from the respective glider deployments, which from Table 3, do not seem to be included in data the set. This needs to be rectified.

> *This comment raises several valid points, which we will address in turn.*
>
> *Firstly, the issue of access to the dataset. The authors apologise for the broken link, which we believe is due to a line-split URL in the type-setting process. All links in the paper have been revisited and reformatted with shorter DOI links to prevent this from reoccurring. All links are now valid.*
>
> *Secondly, we have clarified the contents of the output files in the manuscript in the Data Provenance and Structure section.*
>
> *Lastly, regarding the temperature and salinity data; these were deliberately omitted as they already reside in pre-existing EGO format netCDF data files from the AlterEco missions, and we wish to avoid duplication. This has also been clarified in text in the Data Provenance and Structure section, with associated links to the primary data in Table 1.*

Below I have provided a number of general comments and in some cases queries that require clarification or could be better addressed in the text (to prevent others from being uncertain).

The first and most prominent point being that this publication represents a data set and with the accompanying text, showcases the processing steps. But to me, the publication should be representing both the processor itself that was used to generate the data set as much as (and perhaps even more so) than the data set itself. The ability of this processor to be run in delayed mode as well as both near real time and to be able to switch between different input data files (based on their availability) and to choose between different quenching correction approaches is something that could definitely benefit the broader scientific community. I would highly recommend that this processor be published alongside the data set if at all possible. If so, then I think that the title needs to better reflect the publication of both the data set and the processor, even if links to the processor are published elsewhere but referenced within the text.

*We agree with the reviewer entirely, and it has always been our intention to publish the processor. At the time of publishing the original manuscript, the processor was not in a state to be shared with the glider community. However, we have made substantial progress in this regard during the review process and the URL where we expect to publish the methodology is now included in the Code and Data Availability section of the manuscript.*

Application of a new net primary production methodology; a daily to annual-scale data set for the North Sea, derived from autonomous underwater gliders and satellite Earth observation

**More detailed comments:**

Line 5: Here we introduce a powerful new technique

To be fair, the publication is actually presenting the data set (not the technique as such..... although as mentioned above I think that it should be both). In addition, the technique of merging in situ glider profiles and satellite Earth Observation measurements to produce NPP is not unique. You are not the first (and unlikely to be the last) to find oneself in a situation where a PAR sensor is absent from a glider (or float) deployment and thus has to resort to satellite PAR to run an NPP algorithm. I appreciate that your exact processor as a whole might be unique but the approach is not especially. This does not however detract from the value of the data set. I simply think that the uniqueness of the approach should not be overemphasised.

*The reviewer makes two good points here, the first being that the paper focussed predominantly on the data set and not the methodology, the second concerning overemphasis. We have adapted the text as follows;*

*Here, we introduce a new data set generated using a technique based on the synergistic use of in situ glider profiles and satellite Earth Observation measurements which can be implemented in a real-time or delayed mode system. We apply this system to a fleet of gliders successively deployed over a 19-month time-frame in the North Sea, generating an unprecedented fine scale time-series of NPP in the region (Loveday and Smyth, 2020}. At the large-scale, this time-series gives close agreement with existing satellite-based estimates of NPP for the region and previous in situ estimates. What has not been elucidated before is the high-frequency, small-scale, depth-resolved variability associated with bloom phenology, mesoscale phenomena and mixed layer dynamics.*

Line 8: (Loveday and Smyth, 2020)

I tend to prefer not including references in an Abstract if possible, but perhaps in this instance it may be necessary. I would nonetheless appreciate a few words that elaborate on what that particular study was about? That being said, I tried to look up the reference and it appears to be a link to a published data set which I was unable to access (i.e. not a research publication), which made me wonder how is that published data set with BODC different to the one being proposed to be published here with ESSD?

*We agree that this is little ambiguous. ESSD requires that data sets are published in the public domain prior to the writing of data papers about them. Loveday and Smyth (2020) is the reference given to the data itself, as published by the British Oceanographic Data Centre, the holder of the data. This includes all metadata associated with the mission traceability and data provenance. The ESSD manuscript under review here is about the generation of this data.*

*We have, however, removed the reference from the abstract as the link to the data is better presented in the context of the manuscript itself.*

Line 9 and elsewhere in situ: Maybe I am wrong but I thought in situ had to be in italics

*This has been corrected throughout*

Line 10; "bloom phenology, mesoscale phenomena and mixed layer dynamics"

Has someone from the programme already used this data set to investigate any of these processes? If so, please elaborate so that a reader knows what the original data set was used for (i.e.primary research questions and findings).

*Not as yet, this manuscript is one of the first to be published from the programme. Future publications, based on the source data, are expected to cover mesoscale phenomena and mixed layer dynamics. Future publications based on the data presented here are expected to cover bloom phenology.*

Line 20 " This is in contrast to the international Argo float programme"

I am fairly certain that you are wanting to highlight the operational reach of Argo floats relative to ships, but since the previous sentence starts with " ...the adoption of autonomous platforms has improved the operational reach of traditional research vessels" and this sentence starts with "This is in contrast to the international Argo float programme" it feels as though you are contrasting autonomous platforms with argo floats which I don't think was your intention. Perhaps this can be slightly reworded.

*Thank you, this has been adapted as follows: "Alongside this, the international Argo float programme has grown from zero to over 4,000 floats in a little over twenty years"*

Line 33: Please include a brief description of additional limitations of satellites to correctly interpret ocean colour in coastal regions due to heavy sediment load, shallow bathymetry, super high chlorophyll that is masked out of standard algorithms etc.

*The following sentences have been added:*

*The coastal domain also presents specific challenges for remote sensing of ocean colour in particular. Strong scattering, associated with high sediment loads, and absorption due to non-algal material and CDOM, make chlorophyll retrievals in Case-2 waters challenging (Morel et al., 2006; Sathyendranath et al., 2000) This complexity is compounded by the effects of bottom reflectance from shallow bathymetry (e.g. Ohde and Siegal, 2001) and chlorophyll signals that may be too high to be interpreted by standard algorithms, resulting in excessive masking.*

Line 55-56 " For the first time we uncover the considerable temporal and spatial variability in NPP,"

This is not the first time that anyone has uncovered considerable temporal and spatial variability in NPP from a glider deployment. Maybe downplay the novelty of the approach a bit... it does not detract from the novelty of the specific dataset being generated and its potential research impact.

*We acknowledge that this sentence was far too general for this level of hubris! We have toned down the language extensively and apologise to the glider community at large. Please see the following point for adaptation of this paragraph.*

Line 56-57: driven by seasonal succession, fronts (Miller, 2009) and topographical features,

You do not provide any references to back up the statement that the temporal and spatial variability in NPP observed in your specific data set can be attributed to fronts and topographical features. If this has been done by others then references are required. Alternatively, rather highlight that this is how the data set can be used by others, which will further showcase it's value to the community (but as mentioned before, please ensure that temperature and salinity are included so that the role of physical processes such as fronts, stratification, mixed layer dynamics can be investigated in tendem with the NPP).

*Thank you for this point. We note that the point about temperature and salinity data has been dealt with later in the response to the reviewer. We have re-worked this paragraph extensively as follows:*

*We uncover the considerable regional temporal and spatial variability in NPP across this region, capturing two winter seasons which are crucial in conditioning the system for the following spring and summer periods. We expect future analysis of this data set, the first of its kind for the region, to provide new insights into the biophysical interplay between NPP and a complex regional oceanography defined by the influences of strong tides, topography and fronts (Miller, 2009; Huthnance, 1991). The data set is made available via the British Oceanographic Data Centre (BODC), under [https://doi.org/10/fm39](https://doi.org/10/fm39) (Loveday and Smyth, 2020).*

Line 64 "back-scattering": figure 1 legend refers to backscattering (no hyphen) and throughout the text it interchanges with backscatter (best to be consistent).

*The consistent term "backscatter" has now been adopted throughout.*

Line 65 and 66: Measurements plural.

*This has been corrected*

Line 67 figure 1: Maybe not necessary, but perhaps fluorescence sensor is more accurate than chla sensor in figure legend?

*This has been corrected*

Line 69: "on to". Entirely possible I am wrong here but should it not be onto (i.e. one word)

*This has been corrected*

Line 76 NPP processor: I like the concept of the "NPP processor" in essence being the sum of all the code that defines figure 2. Perhaps the first time that you use the description "NPP processor' you could explain that that is what you are referring to?

*Thank you, this has been clarified in the text.*

Line 103 " Here, when required, SEO-based PAR data is used in lieu of in situ measurements.'

This sentence caused concern when I first read it as I have dealt with instances in the Southern Ocean where a simple swap out of satellite PAR for in situ PAR would not be possible as the differences in derived NPP were too different. Although I have since read further and realised that you have dealt with it in more detail later, I nonetheless think that this should be mentioned here, so that others like me won't preempt unnecessary concern.

> *We thank the reviewer for this comment. A note pointing to later sections in the manuscript has been added in the sentence in question to allay future readers fears.*

Line 134 " Mixed layer depth (MLD) is then calculated from the density gradient according to Holte and Talley (2009)"

Please elaborate on criteria (i.e. how is it different from de Boyer Montégut)

> *The de Boyer Montegut approach uses a fixed threshold to determine the mixed layer depth. Conversely, the Holte and Talley (2009) approach, originally developed for Argo floats, utilises the hybrid algorithm that takes account of dynamic thresholds as well as the shape of the density and temperature profiles. In the authors experience, this method has far better efficacy in shelf regions as it does not rely on a surface-reference threshold value and is able to discriminate between seasonal thermoclines and the transient, wind-driven, mixing events that are often more biologically relevant.*

> *The text has been expanded upon to briefly summarise the above.*

Line 135 "If the MLD calculation fails, the MLD from the previous profile is used.'

Is there a limit on this ... i.e. for how many profiles in a row can the MLD calculation fail and it will simply use the last measurable one? Also why/ when does the MLD calculation fail?

> *Yes, the limit is 1. This is now reflected in the text.*

Line 138-141

I would suggest rather deleting these 4 sentences from here and instead retaining all PAR related processing to section 3.4.1 and all chla related processing to the relevant section on chla processing.

> *Thank you for the suggestion. However, we have decided not to make this change as moving the section conflates the treatment of the satellite-based PAR data with the glider-based PAR data.*

Line 138 " The in situ [Chl-a] data is similarly treated."

Rather specify that it is converted from fluorescens in units of volts to units of chla by multiplying by the scale factor (calibration coefficient) specific to the sensor and subtracting the manufacturer provided dark count.

> *This has been adapted as suggested.*

Line 139 "In the latter case, [Chl-a] data is discarded where the calibrated value exceeds 1e5 mg m−3

Am I understanding this correctly .... I.e. it is discarded when chla is > 100 000 mg m-3? Surely that can't be right... why would it ever be that high?

> *We apologise, this was, in part, a typographical error. The processor uses a maximum value of 1e3 mg m−3. These values occur very rarely as occasional spikes, and in only one glider (Humpback). The measurements are assumed to be erroneous, and not a reflection of a reliable measurement. Therefore, they are removed. This is now corrected in text.*

Line 140 "In the case of DM processing, post mission calibration factors can also optionally be applied, though none were applied in this Case."

In my experience this can pose a really big problem and should be discussed and addressed in a bit more detail e.g. consider maybe matching glider chla to sat chla to show that the manufacturer provided calibration coefficients per glider fluorescence sensor generated a chla concentration that in general matched satellite within some error range (being cognisant of expected deviation from spatial averaging of satellite versus glider etc.). I have experienced instances where I would simply not have been able to merge two gliders in a time series if I had had to rely purely on manufacturer coefficients (luckily I had in situ chla to calibrate individual glider sensors). If that doesn;t work then maybe show the similarities in chla between gliders for all swap over instances. I.e. that there are no step changes evident in chla profiles that could be the result of an unsuitable manufacture calibration coefficient of the sensor.

> *The reviewer makes many good points here. Although our method can accommodate calibration factors from field measurements, unfortunately, no independent measurements of chlorophyll concentration (either fluorescence or HPLC) were taken during the AlterEco glider deployments or retrievals. Consequently, we have adopted the first approach suggest by the reviewer and have compared the quenching corrected glider surface chlorophyll measurements with those from the CMEMS 4km daily reprocessed Level-3 (OCEANCOLOUR_GLO_CHL_L3_NRT) satellite chlorophyll record. This is included as a new figure in the paper (Figure 3). A new paragraph is also included at the end of the "Pre-processing and calibration" subsection.*

Line 149 sub-surface : I think that near-surface or just below the surface would be a better term to use than sub- surface (which implies any depth below the surface and not close to the surface).

> *This has been corrected*

Line 150-151 " using the method described in Hemsley et al. (2015)"

Not really a method.... More so an equation. I would suggest to rather say equation and provide the equation (from Hensley equation 2) and insert it as your equation 1. Then provide detail on what value you used for irradiance reflectance R.

> *This has been corrected as suggested.*

Line 151 " Fresnel reflectance is"

I see Hensley et al (2015) used a set value of 0.48. Please can you elaborate as to why you instead calculated it (as opposed to using standard value as in Hensley et al., 2015) and how your values of

Fresnel reflectance derived from wind speed, relative humidity and mean sea level pressure compared to 0.48? And what the actual equation is to calculate Fresnel reflectance?

> *We thank the reviewer for picking up this point, as it was in fact written ambiguously and incorrectly. We do in fact use a value of 0.48 for the Fresnel reflectance, consistent with Hemsley et al., (2015). The wind speed, relative humidity and mean sea level pressure are used in the calculation of $r_{tot}$ and not in the Fresnel calculations as previously stated. This "Determining the PAR profile" section has now been updated and clarified with the required detail.*

Line 154 "in the previous step"

The previous step does not actually explain how to get E0-. Rather it explains that the method of Hensley et al., 2015 was used to derive E0+ from E0-. I think better to be specific.

> *The reviewer is correct, this has now been clarified in text.*

Line 154-155 " SEO KdPAR , calculated from SEO Kd490 Saulquin et al. (2013), is then used to project broadband PAR into the subsurface across the glider depth record.'

This step is not clear to me.

Saulquin et al. (2013) states that existing models calibrated in open ocean waters tend to underestimate the attenuation of light in coastal waters. They investigate two relationships between KdPAR and Kd490 for clear and turbid waters using MERIS reflectances and the spectral diffuse attenuation coefficient Kd($\lambda$) developed by Lee (2005).

Please can you clarify which equation you used to determine KdPAR from SEO Kd490?

> *We apologise for the confusion in this section, it has been largely re-written for clarity and affects many of the points below. As part of this revision we have clarified that derivation of KD_PAR from KD_490 using equation9a dna 9b from Saulquin et al., 2013.*

Line 156 "Where SEO Kd490 is not available"

Why would SEO Kd490 not be available if you have SEO PAR? Are they not produced by the same satellite product?

Is it possible to compare / provide some indication of the comparison of KdPAR calculated from KD490 (when you have it) and KDPAR calculated from chla? Do they provide similar answers in KdPAR? Are they interchangeable? If different, how much does it affect final NPP calculations? Which is better? Why?

It is entirely possible that I have completely misunderstood this step but if that is the case then it is possible that others will too, so please explain with a bit more clarity. Thanks.

> *SEO Kd490 and SEO PAR are indeed derived from the same source, and yet sometimes the former is not available when the latter is. We presume this is due to the fact that the PAR product is typically derived "in atmosphere", whereas the Kd490 product requires information from the in-water bio-optical model, which may fail due to various reasons. That said, we have not done an in depth analysis to confirm this, as this falls outside of the scope of this paper.*

*Similarly, while an in depth comparison of Kd490 vs KdPAR, as derived through different methods would be a worthwhile analysis, we suggest that this would be future work. It is noted as an avenue for improvement in the "Limitations, scope and future improvements" section at the end of the article.*

*We have, however, made some clarifications to our methodology in the text to clarify the initial point.*

Line 157 "PAR(0)"

Do you instead mean PAR E0+ as described in the section immediately above? Same comment applies to equation (1).

*Here we mean PAR at the surface, e.g. Z=0. The nomenclature has been clarified in the text*

Line 159-169

Should equation 2 that defines Zeu not come before equation 1 that utilises Zeu?

*Thank you for pointing this out, we have rearranged the equation order.*

Line 161-162: ii) substantial gaps in the [Chl-a] data,

Not sure I understand this completely. Assuming this is not relevant to PAR from glider but only PAR from SEO? Even so, this is I think only relevant to instances where SEO Kd490 is not available as only then is chla used in the calculation of Zeu, which is then used to generate KdPAR? More clarity required please. More detail is also required on what threshold constituted "substantial gaps"?

*This point relates to that given in line 183 – which we refer to reviewer to for more information. The text on the interpolation of the backscatter and CHL-a profiles onto the depth record have been moved to the "Pre-processing and calibration" section, as they then provide the context required to interpret the expansion on the text referred to above.*

Line 163-164;

The latter criteria prevents the glider from deriving NPP estimates from [Chl-a] readings that may have been gathered at depths where particle re-suspension in likely to make them unreliable.

Change in likely to is likely

*This has been changed*

Why is this exactly? I realise that you would get enhanced scattering from sediment load but why would that drive incorrect fluorescence readings? Does shallow topography tend to overestimate fluorescence? Or are you anticipating viable chlorophyll to be present in the sediments that were resuspended e.g. from dissolved organic matter?

*This was rather ambiguously written. We did not mean to infer that the [Chl-a] record was likely to be poor in these circumstances. Rather we hoped to convey that the light field in these circumstances is likely to be more complex than we can interpret, and that any quenching*

*methods that rely on the use of backscatter are likely to give poor quality results. We have re-written this paragraph to better explain ourselves.*

165-166: Euphotic depth (Zeu : defined as the 1% of the surface light depth) is subsequently calculated for all good profiles, and is selectively used in the correction of the [Chl-a] profile.

Delete "the" in "defined as the 1%" i.e. to read defined as 1%

*This is corrected, thank you.*

Also, Zeu is not 1% of the surface light depth, but rather the depth at which PAR is 1% of surface PAR. It is also a bit confusing that Zeu is defined as both the depth at which PAR is 1% of surface light (i.e. 1% of PAR E0- or is it PAR E0+?) and as 34 x chl^0.39 in equation 2. Do you use one approach for glider PAR and the other for SEO PAR?

*We have clarified the definition of Zeu.*

Please clarify Zeu being used in the correction of the chla profile to being used in some quenching correction approaches (discussed further in section 3.4.3). Indeed, the Xing et al., method that you ended up using for your data set being published here did not require Zeu for the correction of the chla profile.

*This has been clarified in the text. It is noted that Zeu is also not required in all cases, e.g. for the Xing et al method.*

168: Ed.

I believe that Ed is the incorrect term in this instance as you are comparing surface PAR (defined here as E0+ and not Ed) as per y axis title in figure 3 a) and b).

*Thank you for pointing this out, it has been corrected*

169: ~50 W m-2

How did you calculate this? Is this simply the average midday surface PAR of Glider minus MODIS? Also, please clarify what a difference of ~50 W m-2 amounts to in terms of %. For example, a typical underestimate of ~50 W m-2 from MODIS derived surface PAR relative to a glider surface PAR of 250 W m-2 is a 20% reduction, which to me feels substantial rather than insignificant.

If the MODIS surface PAR typically underestimates glider surface PAR is it not best to try to correct the MODIS derived product e.g. by applying a % increase to surface PAR generated from a regression between the two products?

I was wondering whether perhaps a good comparison to accentuate the similarities/ differences between the two methods would be to compare daily integrated PAR as this is in essence what is used in the NPP model?

Strange that for 454 Cabot the typical reduction in midday MODIS surface PAR relative to glider surface PAR was not evident in August. Any idea why this would be? As opposed to a more consistent midday offset for the rest of the timeseries?

I am particularly interested in these results because, as mentioned earlier, we have tried this approach in the past but had little success primarily because the offset between glider and satellite PAR was too large and inconsistent. I am trying to remember if perhaps it was because I only used midday PAR for daily NPP calculations? It would be great to be able to better understand why the differences were inconsequential in your case and how this approach may be better implemented in other ocean regions and platforms (e.g. BGC- Argo).

> *There are a number of points to clarify here. We will visit each in turn.*
>
> *Firstly, as the reviewers suggests, this estimate of ~50 W.m-2 was calculated as a simple average difference of glider vs MODIS at midday. However, on revisiting our analysis, this seems to have been a rather unfair and biased comparison, especially as the midday value does not always correspond to the PAR maximum in the glider PAR record. As a result, we have revisited our analysis, and made the following updates.*
>
> - *The original statistics on Figure 4 remain as they give a good overview of the correlation between the glider and SEO daily instantaneous PAR records.*
> - *However, at the reviewer's suggestion, we have included a new pair of black traces on each panel of the figure, showing the integrated daily PAR values for the glider (solid black line) and SEO (dashed black line).*
> - *As the daily integrated value is the critical quantity for measuring NPP, the 50 W.m.-2 value is replaced by the discussion of the mean daily integrated PAR value, which compares much more favourably between the SEO and glider-based PAR. We thank the reviewer very much for this suggestion.*
>
> *Secondly, suggestion to correct the SEO PAR with glider PAR, where available is a good one. However, in this case, as we do not have glider-based PAR across an entire season, we are concerned that introducing such a calibration may lead to a summer bias. We do note, however, that when gliders bearing PAR sensors are regularly deployed in a region, it may be advantageous to compare their surface signals with satellite products with a longer-term view to building up a more comprehensive picture of how to inter-calibrate between the two.*

171-172: cycle, with a mean̄ E+o that falls within 7% of the in situ broadband value.

What exactly is meant by broadband value?

> *The term "broadband" has been removed, as we refer to Eo+ in both cases.*

172: that: Delete that

> *This has been corrected*

174: 3.4.2.

Using the optical backscatter profile back-scatter, backscatter, backscattering, please be consistent. My personal preference is for 'backscatter'

> *All instances have been changed to backscatter throughout.*

What about dark correcting? Was this done? By subtracting both the manufacturer provided dark count and presumably an in situ derived dark count?

*Please see the comment on line 181-182*

180: with measurements of [Chl-a] < 0.0 mg m−3 discarded.

Could be that your dark count was too high?

*Please see the comment on line 181-182*

181-182: To account for bio-fouling, the record is also discarded where there is a consistent "step change" of > 5 mg m−3 in [Chl-a] at depths below both the MLD and Zeu , as compared with the initial deployment value.

Interesting way to account for biofouling. I have never come across this. My first question would be why would you expect biofouling to generate a step change? My understanding is that biofouling would be gradual and reflected in a drift in the in situ derived dark count, which can subsequently be used to correct for biofouling (to a point).

I see no mention of the dark correction being applied to the glider fluorescence profiles, this should be specified in the section which notes the application of calibration coefficients. Over and above the manufacturer derived dark correction an in situ derived dark should be applied (to account for ageing of sensor drift or biofouling or change in dark current when integrating sensors onto gliders). This can be generated from each glider mission in delayed mode (if no drift is evident during the deployment e.g. from biofouling) or per profile (which can be implemented in near real time). This is typically done by generating a per profile minimum (see Wojtasiewicz et al. 2018).

Note however that the fluorescence minimum should only be estimated for depths below the Zeu where quenching is not occuring and that this approach can be tricky in shallow low oxygen waters where fluorescence can increase rather than decrease at depth due to DOM.

*This point is in response to the comments made on lines 174, 180 and 181-182. There are a number of issues to address here.*

*Firstly, the issue of dark calibration of both the fluorescence and backscatter profiles. We completely omitted this information from the original manuscript, which was an oversight to say the least! In addition, our previous statement of "with measurements of [Chl-a] < 0.0 mg m−3 discarded" is quite misleading. We have added substantially to the "Pre-processing" (now "Pre-processing and calibration") section to deal with both backscatter and fluorescence dark calibration. All information relating to calibration has been removed from the "Quality control and quenching correction of the chlorophyll fluorescence profile" section, which has been renamed "Quenching correction of the chlorophyll fluorescence profile. This information is now included in the "Pre-processing and calibration" section.*

*Secondly, on the point about backscattering. Toward the end of its sampling, glider Kelvin experienced a sudden shift in the recorded fluorescence values, equating to a "step change" in [Chl-a] of 5 mg.m3. We attributed this to biofouling, but perhaps this was an unjustified leap. As the reviewer points out, this may be the result of sudden shift in the performance of the sensor. Our method was designed to discard data the erroneous data, but it is not an indication of biofouling per se. Consequently, we have removed the reference to biofouling and explained the situation more explicitly.*

*Lastly, the dark correction approach we used was only applicable to delayed mode processing. We thank the reviewer for pointing us to Wojtasiewicz et al. 2018, work that we were not aware of, but which is now included in the future work section as a proposed improvement to the processor.*

183: Where the interpolation fails due to lack of data

What defines a failure? Trying to think why there would be a scenario where there was not enough fluorescence data within a single profile to facilitate a depth interpolation?

*As the CTD and [Chl-a] sensor do not record data at the same frequency, very short dives may occasionally lead to insufficient data to perform the interpolation. In addition, EGO format glider data is subject to various quality control measures, the extent and nature of these are determined by the original data owner. These QC processes occasionally remove sections of the fluorescence data, making interpolation impossible. We would like to point out that this is a rare occurrence.*

*Discussion of interpolation has now been moved to the "Pre-processing and calibration" section, where the above arguments are briefly included.*

196: – The Swart et al. (2015) method

It is impossible for me not to ask this, but why didn't you rather test the Thomalla et al., 2018 method? The Thomalla et al., 2018 method was an improvement based on the Swart et al., 2015 method, which we would never have developed had the Swart et al. (2015) method (or any of the others for that matter) worked for that particular glider time series which displayed a significant subsurface chla maxima. In addition, the 2018 paper shows that none of the other methods (Xing, Biermann, Swart or Hemsley) worked well for that glider time series highlighting it as an apporporiate option to have compared in your processor. The key approach that the Thomalla method takes, which the others do not, is to try to figure out the actual depth range of quenching where the correction needs to be applied.

That being said, given how well the Xing method is shown to perform for your glider time series together with your description of the failure of the Swart and Hemsley methods that rely on backscatter, it would not surprise me if the Thomalla et al 2018 did not perform any better. Nonetheless, I would prefer to see it rather than or in addition to the Swart method (since it is an upgrade on a similar concept) and even if you choose not to include the Thomalla method I would very much like to see an additional panel included in Figure 4 to show the comparative performance of the Swart method.

*Here, we owe the reviewer an apology. The NPP processor has been some years in development and the Thomalla et al., (2018) was not available when the system was initially designed. In addition, as not all of the gliders in the AlterEco programme were deployed with backscatter sensors, it would not have been possible to apply the Thomalla et al. (2018) method ubiquitously. In practice, however, the Thomalla et al. (2018) could have been used for gliders with the required payload. We have included statements to this effect in the conclusions of the paper, marking the inclusion of updated quenching methods as a priority in future development of the processor.*

*Further, we have included the performance of the Swart et al., (2015) in the results on figure 4. As expected by the reviewer, it does not perform well in this case. We suggest that this is related to the prevalence of relative high concentrations of sediment in the North Sea, which undermines the assumption that backscatter is mostly associated with the presence of [Chl-a] – something that is likely true in the open ocean.*

209-215: The Xing et al. (2012) method clearly outperforms the other methods tested

I think it would be good to include a description of why and when this method has been shown to fail in other circumstances e.g. when chla max is below MLD but above Ed. This information is especially useful if the processor is published and other users have the option to implement the different quenching options.

*We are unsure of the reviewers point here. In the paper we suggest that the Xing et al., (2012) method works well precisely because it does not project a deep chlorophyll maximum that is below the MLD to the surface.*

213-215: In this case, quenching corrections using euphotic depth as a maximum depth limit (e.g. (Biermann et al., 2015)) over-correct as they tend to encapsulate the DCM in the quenching correction process, extrapolating erroneously high [Chl-a] to the surface.

Perhaps worth including a sentence to explain that this approach was indeed designed to account for sub surface chla maxima but in open ocean regions where the MLD is typically deeper than the Ed.

*This is included.*

218: for each using the solar irradiance: For each glider profile using the solar irradiance model

*This has been corrected*

219: the model runs using the [O3]. What does the [O3] mean?

*[O3] refers to the total column atmospheric ozone. This has been clarified in the text.*

231-232: The processor retains the ability to implement this method, as detailed extensively in Hemsley et al. (2015) and represented in Figure 2 by the "Case 1" decision box.

Similar comment as previously regarding the fact that statements such as this are referring to the processor itself but not the data set being published here.

*We have noted this but have not made any specific changes relating to this comment.*

238: calculated from the corrected [Chl-a] and depth profiles and spectral downwelling PAR

Needs to be revised e.g: calculated from the corrected [Chl-a] and spectral downwelling PAR profiles

*This has been adapted as the reviewer suggested*

244: uncorrected

As in raw uncalibrated quenched profile?

Why is NPP calculated for the uncorrected chla profile?
Line 239 says that Net primary production is calculated from the corrected [Chl-a] profiles only.

*Sorry, this should have read as "corrected profile" only. It has been fixed in the text.*

248: subsequently: I think should rather be subsequent

*This has been corrected*

248: Figure 7

I have to say I am amazed at the relatively miniscule difference in daily integrated NPP given the relatively big difference in midday PAR between the two methods. We tried a similar approach in the Southern Ocean with a different NPP model but our results were not comparable and we had to adjust Satellite PAR to get similar rates of NPP.

The periodic drops in daily integrated NPP to zero for both methods when no data is available (e.g. on 4 occasions during 454 Cabot) is i think not real. Perhaps you can rectify this in the presentation of the time series either by interpolating between profiles or rather just have gaps in the time series for where there is no data (rather than periods of a presumably "fake" zero NPP).

*Thank you for pointing this out, the figure has been updated as suggested*

252-253: Two peaks are captured in April/May 2018 and April 2019, corresponding with the timing of the regional spring bloom, and with the latter event significantly more intense.

I am simply not seeing this?

I don;t see a double peak in April/ May 2018... maybe May/June 2018 if I try hard? But more likely what I would consider a highly variable system and not a two peak (double bloom) phenology. Similarly I do not see 2 peaks in April 2019... maybe a second peak in May but certainly not one that is significantly more intense.

*This section has been re-written in response to the points above. Reference to double peaks has been removed in favour of discussion about the expected seasonal changes and variability in general.*

253-255: Independent [Chl-a] estimates, derived from v4.2 of the ESA Ocean Colour Climate Change Initative (OC-CCI) data set (Sathyendranath et al., 2019), indicate that the 2018 spring bloom was relatively intense, suggesting that 494 (Stella) failed to fully capture this event.

Initiative spelt incorrectly

*This has been corrected*

You need to show this data somewhere i.,e. Compare sat chla and glider chla for the entire time series and then discuss further, highlighting spatial / temporal averaging etc.

*This information is now included on figure 9. The light orange trace shows the NPP as extracted from the v4.2 OC-CCI NPP product. This is in only available until the end of 2018, so does not*

*cover the entire glider time-series, however it does demonstrate the point we make in the text. The final paragraph of the "Calculating NPP" section has been updated to discuss this further.*

*Much of the discussion of figure 9 has now been moved to paragraph two of the, now expanded, "Comparison with historical measurements in the North Sea" section, as it is more appropriate here.*

257-259: However, the divergence between the signals recorded by the concurrently deployed gliders 455 (Orca), 497 (Humpback) and 454 (Cabot) strongly suggests the presence of significant spatial heterogeneity in the region north of the Dogger Bank.

Where exactly is the Dogger Bank? can you maybe identify it on Figure 1? It is hard for me to be able to isolate evidence of significant spatial heterogeneity in the region north of the Dogger Bank when all the gliders seem to sample a similar region but evidence of spatial heterogeneity appears to be only in May-July. Perhaps this is more suggestive of seasonal heterogeneity rather than regional? Notably 478 Eltanin and 477 Dolomite covered very different regional domains but nonetheless had very similar NPP possibly providing evidence of the absence of patchiness and a more homogenous system at this point in time (phase of seasonal cycle)?

*The location of the Dogger Bank is now clarified on Figure 1 and referred to in text. The text has also been adapted slightly, as further analysis of this point is left to future work.*

275: (1) bloom growth or decay

Decay refers to only one loss term e.g. does not include grazing, maybe needs a term that encompasses all e.g. bloom growth or loss (e.g. respiration, decay, grazing)?

*We thank the reviewer for pointing this out. The text has been updated to "bloom growth or loss (e.g. via respiration, decay, grazing).*

277: exceeds 5 mg m-2

More like 4 mg m-2 I would say?

*This has been adapted as the reviewer suggests*

278: night-time cases uncorrected cases is

Delete one of the cases

*This has been corrected*

296-297: The annual cycle and mean annual NPP rate agrees well with contemporaneous values interpolated from the monthly mean OC-CCI based NPP climatology for 1998-2018 (Kulk et al., 2020).

What NPP model was used for satellite derived NPP?

Were those for the same region, can you provide details on spatial averaging? What were the actual numbers compared to yours? Why not rather compare your results with satellite derived NPP for the same time period (and spatial region) as the glider deployments?

*This section of the manuscript has been updated extensively to address the issues above. It now read as follows:*

*Kulk el at. (2020) applied the NPP methodology of Platt and Sathyendranath (1988) and Sathyendranath et al. (2020) to the OC-CCI ocean colour record, producing a 20 year time series of satellite-based NPP from 1998-2018. This record does not span the entire AlterEco period (2017-2019), is only calculated at 9 km resolution, and is only available on a monthly timebase, so we are unable to make a direct comparison with the glider data. Instead, the monthly mean and standard deviations for the are calculated from the climatology across a box spanning the AlterEco sampling region. The results are shown in the final two columns of table 5. The annual cycle and mean annual NPP rate as measured from the glider missions agrees well with contemporaneous values interpolated from the monthly mean OC-CCI NPP record. It is notable from the table that, while still within one standard deviation, the April NPP peak in the glider data is somewhat lower than its OC-CCI counterpart, likely due to the low signal recorded by 494 (Stella) over this period in 2019. However, the glider-based NPP signal peaks at a time consistent with remote sensing estimates.*

298: the April NPP peak in the glider data is somewhat lower than its OC-CCI counterpart, It is not clear to me what figure in Kulk et al., 2021 you are comparing your results to?

*Sorry, this was misleading in the text. The Kulk et al., 2020 reference points to the OC-CCI NPP dataset in general. The statement about the peak is related to table 5. This has been clarified in the text.*

301-303: The glider-based annual mean NPP value 98 gC m−2 a−1 is comparable with the 119 gC m−2 a−1 measured through extensive surveys carried out over ICES Region 7 (north of Dogger Bank) by Joint and Pomroy (1993), as well as with observation based estimates of 125 gCm−2 a−1 for the northern North Sea (van Beusekom and Diel- Christiansen, 1994).

Can you provide more information e.g. method of measuring NPP (14C?). How did they get to annual? measure monthly and integrate?

*The manuscript is adapted to reflect this information as follows;*

*The glider-based annual mean NPP value 98 gC m−2 a−1. This compares favourably with the 119 gC m−2 a−1 measured by Joint and Pomroy (1993), who applied a $C^{14}$ approach to measure daily NPP through extensive surveys carried out over ICES Region 7 (north of Dogger Bank), scaling up to monthly estimates using the mean daily value across the region and number of days per month. It also compares well to the annual estimate of 125 gC m−2 a−1 for the northern North Sea proposed by van Beusekom and Diel- Christiansen (1994), based on their synthesis of daily NPP estimate from multiple cruises.*

309:

Similar modification of the glider based NPP measurements to a GPP estimates How were these done?

*Measurements of NPP and GPP in the North Sea are relatively sparse. In order to compare our NPP measurements with published GPP measurements we make the broad assumption that the ratio of NPP:GPP remains broadly constant on an annual basis across ICES region 4. Using the values presented in Varela et al. (1995), we can then estimate what the glider NPP*

*measurements would approximately translate to in GPP, thus allowing us to compare our findings with Capuzzo et al. (2018). This approach is now clarified in the text.*

315: sky conditions

As in cloudiness? Isn;t this mostly accounted for in your NPP calculations?

*Where we use glider-based PAR, most definitely yes. However, when we use the EO-based PAR from MODIS, the cloudiness is invariably averaged across the day. Consequently, although cloudiness is captured in the mean, we lose any high-frequency variability.*

329: profile

Change to profiles

*Corrected*

360: weeks re-emerge

Change to weeks to re-emerge

*Corrected*

363: an 19-month

I think should be a 19-month

*This has been corrected*

370-371: present in the production in region

Change to.. present in the rates of NPP for the region perhaps?

*Thank you for the suggestion, this has been adopted in the text.*

371: and highlight the advantages of using autonomous systems to persistently monitor the shelf-seas, especially in tandem with remote sensing based approaches.

I think that this could be elaborated further to highlight the value of the data set that has been generated here.